# Long-range optofluidic control with plasmon heating

B. Ciraulo [1,2], J. Garcia-Guirado [1,2], I. de Miguel[1], J. Ortega Arroyo [1,2 ✉] & R. Quidant [1,2,3 ✉]

Using light to manipulate fluids has been a long-sought-after goal for lab-on-a-chip applications to address the size mismatch between bulky external fluid controllers and microfluidic devices. Yet, this goal has remained elusive due to the complexity of thermally driven fluid dynamic phenomena, and the lack of approaches that allow comprehensive multiscale and multiparameter studies. Here, we report an innovative optofluidic platform that fulfills this need by combining digital holographic microscopy with state-of-the-art thermoplasmonics, allowing us to identify the different contributions from thermophoresis, thermoosmosis, convection, and radiation pressure. In our experiments, we demonstrate that a local thermal perturbation at the microscale can lead to mm-scale changes in both the particle and fluid dynamics, thus achieving long-range transport. Furthermore, thanks to a comprehensive parameter study involving sample geometry, temperature increase, light fluence, and size of the heat source, we showcase an integrated and reconfigurable all-optical control strategy for microfluidic devices, thereby opening new frontiers in fluid actuation technology.

[1] ICFO – Institut de Ciències Fotòniques, The Barcelona Institute of Science and Technology, Castelldefels (Barcelona), Spain. [2] Nanophotonic Systems Laboratory, Department of Mechanical and Process Engineering, ETH Zurich, Zurich, Switzerland. [3] Institució Catalana de Recerca i Estudis Avançats (ICREA), Barcelona, Spain. ✉email: jarroyo@ethz.ch; rquidant@ethz.ch

Controlling long-range transport of fluids is fundamentally important to any application that is based on microfluidics, from standard flow cell assays, ubiquitously encountered in biophysics, cell biology, and nanomedicine, to lab-on-a-chip devices. Although pressure-drive control systems and syringe-pumps have become the methods of choice in microfluidics, there is an evident scale mismatch between the fluid system to be manipulated and the controller[1–4]. As a result, there has been significant interest to develop integrated microscale alternatives[5]. Many of which rely on establishing non-equilibrium conditions within the sample, either by chemical[6], electric[7], surface tension[8] or thermal gradients[9] or a combination thereof. Despite extensive efforts, the development of such microscale controllers has been challenging due to the difficulty in reliably establishing the desired non-equilibrium conditions and simultaneously characterizing the effect these have on the fluid and particle dynamics.

A promising approach to achieve integrated fluid manipulation is to use localized thermal gradients[10,11]. As a heat delivery mechanism recent progress in the field of thermoplasmonics[12,13] has shown that nanoscale structures can efficiently transduce light into heat, thereby enabling a high degree of control over the amount and extent of heat deposited to a specific area[14]. More recently, efforts have focused on exploiting these thermal gradients to manipulate or characterize particles within the fluid[15,16]. Thermal gradients, broadly speaking, alter the dynamics of the particles at two distinct levels: at the particle and fluid level, respectively. At the particle level, the motion of objects in solution along or away the thermal gradient, thermophoresis, is determined by their interactions with the solvent and leads mostly to short-range motion[10]. At the fluid level, thermal gradients can induce short-range motion of the particles by thermo-osmotic flow, and long-range motion by either convection or thermo-viscous flow. Notably, a considerable amount of these studies have been performed under conditions that exclusively favor thermophoresis and suppress convection-driven dynamics[17,18], which results in short-ranged control that is comparable in size with the thermal gradient. To enhance the range and magnitude of fluid manipulation, static or dynamic electric fields have been coupled into the system[19–21], the thermal gradient has been established near a liquid-gas interface leading to Marangoni convection[22–24], or a coating of indium-tin-oxide (ITO) has been added to the substrate containing the heat sources to increase convective flow[25]. Despite these advances, a comprehensive multiscale and multiparameter characterization that addresses the complex nature of the different contributing phenomena responsible for mass and fluid transport has been lacking. This stems from the fact that most studies only estimate the underlying temperature landscape by simulations, and the effect of experimental parameters are seldom explored in detail.

In this work, we address this knowledge gap by introducing an optical platform that is capable of: reproducibly pushing the system away from the thermal equilibrium, characterizing the thermal landscape in situ over three dimensions in a label-free manner, and simultaneously extracting the underlying dynamics over the entire imaging volume. Our all-optical approach combines state-of-the-art thermoplasmonics with digital holographic microscopy, which allows us to perform an exhaustive characterization of a model system and identify the main contributing phenomena to the observed dynamics. In addition, we explore different regimes of the model system by varying the sample geometry and orientation. We then harness the results of this comprehensive study to demonstrate that a microscopic thermal perturbation of only a few degrees is enough to manipulate a fluid from the μm- to mm-scale and achieve long-range transport. Finally, we showcase an integrated strategy for all-optical control

of fluids within microfluidic devices that is reconfigurable, perturbs the system minimally, and does so with a fast time response. As such, our work presents a plasmonic-based optofluidic approach to control particle and fluid motion over varying length-scales, which opens the possibility for a new generation of microfluidic technologies and applications.

## Results

**Experimental concept.** To study the non-equilibrium fluid dynamics at the micro-scale in the presence of a localized thermal gradient we use a flow cell system containing a solution of 1.0 μm polystyrene microspheres as tracer particles. The flow cell consists of two microscope cover glasses separated by a silicon spacer as depicted in Fig. 1a. We define the distance separating the cover glasses as chamber height. The surface of one of the glass coverslips is functionalized with a uniformly distributed monolayer of gold nanorods, AuNRs, with dimensions of $13 \times 46$ nm$^2$ (Supplementary Fig. 1). Resonant illumination with a pump beam (780 nm) converts each plasmonic nanoparticle into an efficient light to heat nano-transducer (Fig. 1b). The contribution from each of the AuNRs excited by the pump leads to a steady-state 3D temperature distribution that extends over the resonant

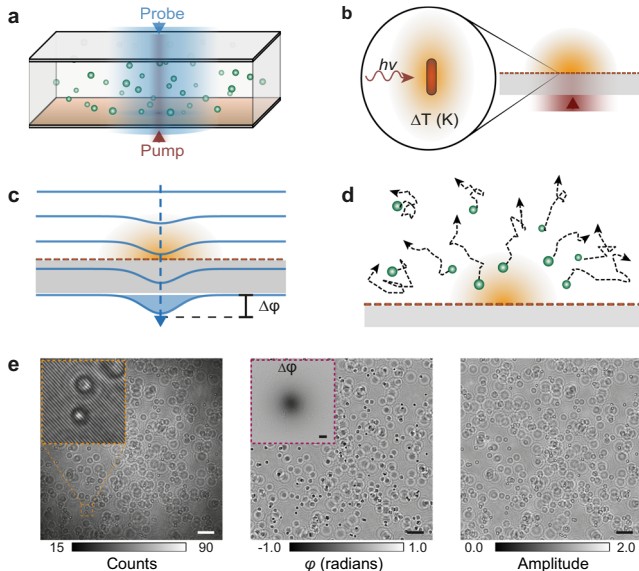

**Fig. 1 Plasmon heating-based optofluidic platform. a** Schematic diagram of the model flow cell system composed of two coverslips separated by a silicon spacer of known size, which is illuminated by counter-propagating pump and probe beams. The flow cell is filled with a solution containing 1.0 μm microspheres as tracer particles and the bottom glass is functionalized with a uniform distribution of gold nanorods to efficiently transduce light into localized heat. **b** Plasmon heating is achieved by resonant illumination of the immobilized AuNRs on the glass surface, resulting in a temperature gradient that extends over the area delimited by the pump beam. **c** The established temperature gradient in the sample leads to an overall change in the optical path length of the incident probe wavefront which is detectable in the form of a phase difference via a wavefront sensor. **d** The dynamics of the fluid in the presence of a localized temperature gradient are studied by following the motion of tracer particles via the principle of single-particle-tracking velocimetry. **e** As an optical readout platform of (**c**) and (**d**) we use a digital holographic microscope, whereby the measured hologram (inset: zoom of the highlighted area) is separated into a phase (inset: measured phase difference caused by heating) and amplitude channel, which report on the temperature and fluid dynamics, respectively. Scale bars: 10 μm.

illumination area[26]; thereby enabling thermal landscape engineering by beam shaping[27,28].

To simultaneously map the temperature field and its effects on the fluid and particle dynamics, we use a probe beam at a wavelength far off-resonance from the plasmonic particles (465/635 nm), and rely on the concepts of wavefront sensing and 3D single-particle-tracking velocimetry, respectively. In brief, our label-free thermometry approach, depicted in Fig. 1c, is based on detecting minute phase changes derived from temperature-dependent refractive index variations in the imaging medium as the incident wavefront travels through the sample, thereby accumulating an overall optical path length difference. Conversely, the dynamics of the fluid are retrieved by following the motion of the tracer particles in three dimensions (Fig. 1d). As an optical platform capable of integrating both wavefront sensing and 3D particle-tracking velocimetry approaches we implemented a custom off-axis digital holographic microscope operating in a pump/probe configuration (Methods). From the measured holograms we extract phase and amplitude images, which we use to measure the temperature profile and the particle dynamics, respectively (Fig. 1e).

**Detection of fluid dynamics by 3D single-particle tracking**. One of the major advantages of digital holography applied to fluid dynamics is digital propagation, which not only extends the depth of focus of the imaging system, but also enables 3D single-particle-tracking over a volume much larger than conventional microscopy[29]. Specifically, we apply the angular spectrum method to generate a stack of images along the optical propagation axis, which we refer to as depth-stack (Methods). Following this notation, we denote the position along the inside of the chamber as either sample depth or channel depth. As shown in Fig. 2a from this depth-stack, every tracer particle within the imaging volume is first segmented, then localized in three dimensions with a precision greater than the diffraction limit, and finally repeating this process over the entire hologram sequence, each particle's motion is reconstructed using established

trajectory linking algorithms (Methods). As a first step, we validate our tracking ability by characterizing the tracer particles at equilibrium conditions (Supplementary Fig. 2).

To demonstrate the principle of particle-tracking-velocimetry we choose a model flow cell system with a nominal chamber height of 50 μm that is orientated perpendicular to the direction of gravity and has been seeded with 1.0 μm tracer particles. Here, we compute the average over the magnitude of the instantaneous velocity, $<|\mathbf{u}|>$, of each tracer and map this value to the corresponding 3D trajectory within the imaging volume. We take the system at thermal equilibrium, where the velocity of the tracer particles is narrowly and homogeneously distributed within the imaging volume, as a reference (Fig. 2b). Upon pushing the system away from thermal equilibrium, the distribution of velocities broadens significantly and shifts to higher values in a position-dependent manner; with higher values for tracks passing in the proximity of the center of the XY plane, and bottom of XZ plane, respectively (Fig. 2c, Supplementary Movie 1). This observation is consistent with the location of the pump beam and by corollary the thermal field, the former with a total cross-section diameter of ~30 μm. Namely, the closer the tracers are to the greatest temperature change, the faster they move (Supplementary Fig. 3). More precisely, the trajectories exhibit dynamics caused by the superposition of Brownian motion with that of convection as expected from systems at low Reynold number[12].

Our optofluidic platform not only induces localized temperature gradients but also measures them in situ, thus enabling us to correlate measured dynamics with a map of the thermal perturbation (Methods). Briefly, this is accomplished by first measuring the optical phase difference between pump On and pump Off cycles (Supplementary Fig. 4). Then, from the detected phase difference, the three-dimensional temperature map is retrieved using established analytical solutions of temperature fields around plasmonic heat sources[30]. Given the micrometric size of the heat source and the low Rayleigh number of our system, conduction is the main heat transport mechanism in the system and as such convection does not significantly affect the steady-state temperature distribution[12]. As an example, we show

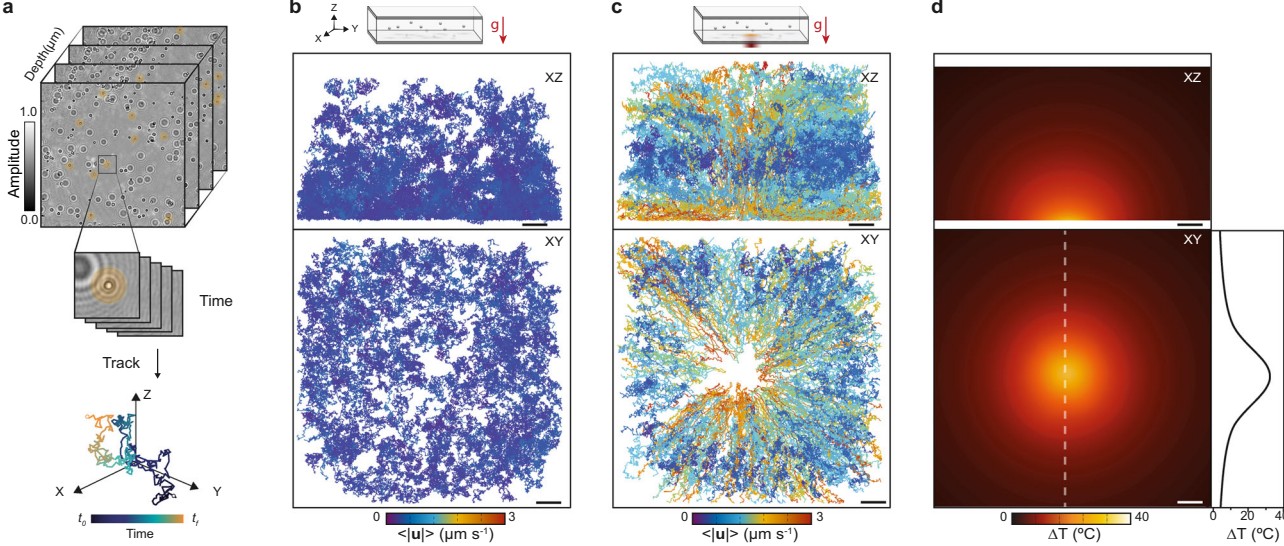

**Fig. 2 Correlative 3D single-particle tracking and label-free thermometry. a** Representative 3D trajectory obtained from a numerically propagated depth-stack of intensity images. **b**, **c** Schematic depicting the orientation of the optofluidic platform with respect to gravity, and spatial distribution map in the XZ (top) and XY (bottom) planes of the trajectories of each tracer particle under thermal equilibrium and non-equilibrium conditions, respectively. Each track is assigned a color according to the average magnitude of the instantaneous velocity. **d** Experimentally measured 3D steady-state temperature distribution due to plasmon heating in the XZ (top) and XY (bottom) planes beside a cross-section along the dashed line, which led to the particle dynamics measured in (**c**). Scale bars: 10 μm.

in Fig. 2d the temperature profile with a peak temperature change of 32 °C with respect to ambient conditions (25 °C), which is responsible for the non-equilibrium dynamics in Fig. 2c. The temperature profile is smooth as expected from a delocalized collective heating mechanism.

**Fluid dynamics characterization and influence of sample orientation with respect to gravity.** Although the trajectories of the individual tracer particles show a clear effect upon thermal perturbation, the motion of these tracers arises by the superposition of different phenomena. On the one hand, there are contributions independent of the thermal gradient, such as radiation pressure, sedimentation, and Brownian diffusion; and

on the other hand, contributions that are driven by the thermal perturbation, such as convective flow, thermo-osmotic flow, thermoviscous flow, and thermophoresis. Despite the superposition of different phenomena, by removing the Brownian contributions and characterizing the radiation pressure and sedimentation terms independently (Supplementary Fig. 5), the tracer dynamics allow us to identify the thermally driven contributions. Moreover, when the fluid transport mechanisms are dominant, the particle trajectories directly report on the fluid dynamics.

We extract the underlying velocity vector field from a set of 3D tracks following the workflow detailed in Fig. 3a. Briefly, the imaging volume is segmented into voxels, and the instantaneous velocity vector, denoted as **u**, is calculated for each segment of

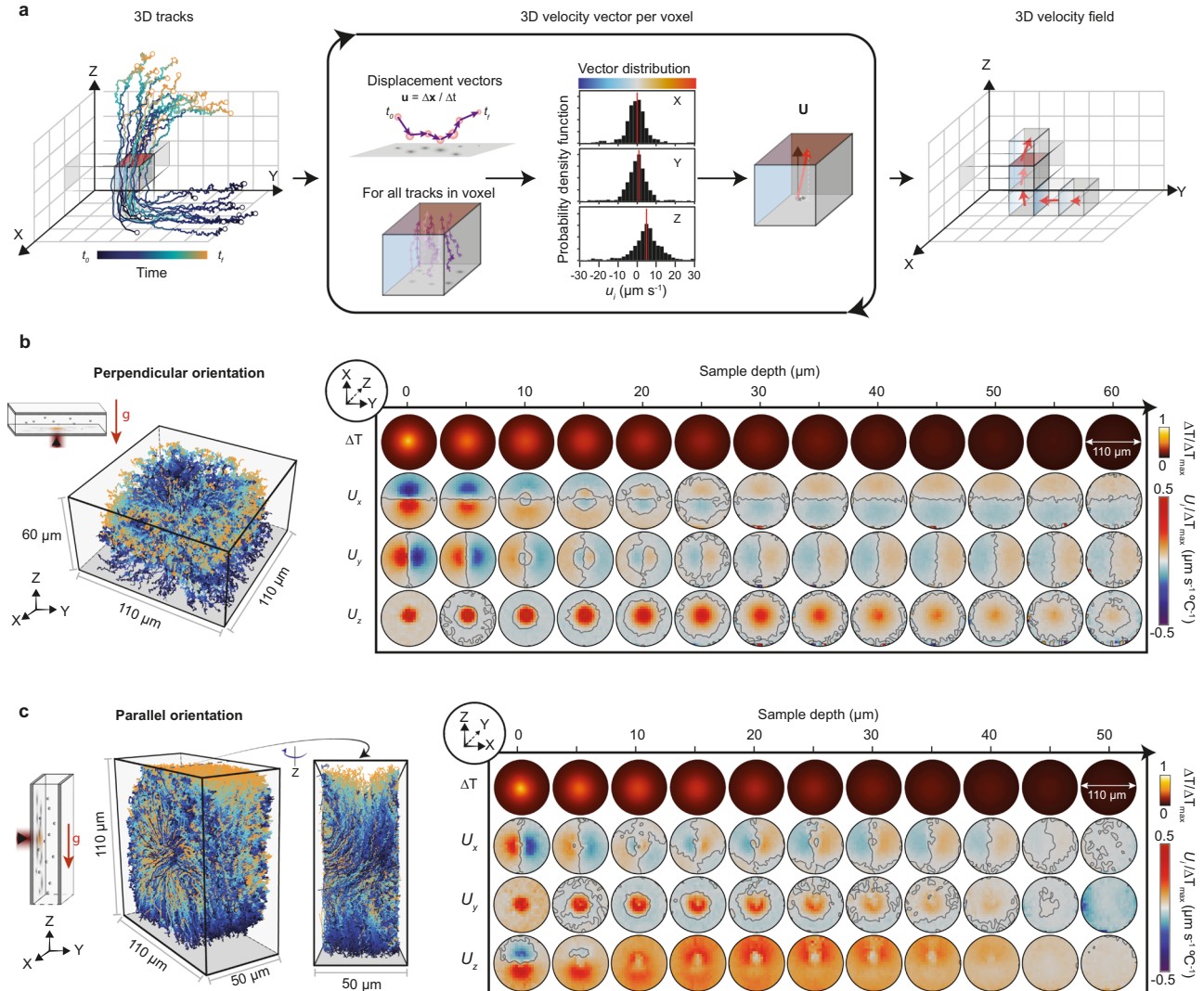

**Fig. 3 Full 3D vector flow characterization and influence of sample orientation. a** Workflow to determine the 3D velocity field over the imaging volume from the measured tracer particle trajectories. First, the instantaneous velocity vectors, **u**, are extracted for every track. Next, the imaging volume is divided into voxels and all instantaneous velocity vectors within a given voxel are collected for subsequent analysis. Then, from the instantaneous velocity vector distribution of said voxel, an ensemble average flow velocity vector, **U**, is determined. This process suppresses the Brownian motion contribution of each tracer particle in **U**. Finally, this procedure is repeated for all the remaining voxels. **b** Representative 3D particle trajectories, with time evolution encoded in color, together with the resulting 3D velocity field map for an optofluidic platform with a nominal channel height of 50 µm oriented perpendicular to the direction of gravity as shown in the schematic diagram. **c** Same as (**b**) but for an optofluidic platform that is oriented parallel to the direction of gravity. The scheme indicates the plane along which the maps in (**b**) and (**c**) are calculated given the change in orientation of the optical axis, XY and XZ respectively. Each row and column in the velocity maps correspond to a different vector component and channel depth position, respectively. Color encodes the direction and magnitude of the flow velocity, whereas the solid lines represent a contour line indicating a change in direction. All velocity vector components share the same magnitude scale, which has been normalized by the maximum temperature increase in the system.

every trajectory. Then, the velocity vector at each voxel, $\mathbf{U}$, is calculated as the ensemble average from the distribution of all $\mathbf{u}_i$ within that voxel. Finally, this process is iterated over all voxels within the imaging volume until a full 3D velocity field map is produced with a resolution determined by the voxel dimensions.

Importantly, the ensemble-based averaging performed at each voxel significantly minimizes the Brownian contribution, since the mean displacement attributed to such dynamics is zero in the limit of many samples. This translates into the need for a high number of tracks, which can be obtained by either seeding the system with a high concentration of tracers, increasing the duration of the experiment, or performing multiple experimental replicas at the same position. Experimentally, on average, we achieve a high sampling number per voxel $<N> = 180$ by a combination of high particle seeding and performing multiple experimental replicas at the same position.

In this work, we show the time-encoded 3D tracks alongside the calculated velocity vector field and the induced temperature field in the form of a two-dimensional array of intensity maps (Fig. 3b–c). Specifically, the rows in the array indicate the specific vector field component ($U_x$, $U_y$, $U_z$), while the columns refer to the position along the optical axis, i.e., the sample depth. The intensity maps encode the sign and magnitude of every component $U_i$ at each position. In detail, we use the same color gradient scheme from blue to red shown in Fig. 3a, to assign the directionality (blue for negative and red for positive) and magnitude (color intensity). Only voxels satisfying a high sampling number, $N > 25$, and a high SNR are shown in each map. The circular-shaped velocity maps are due to the lower probability of finding long tracks at the image boundaries.

We use this approach on our model flow cell with a channel height of 50 μm to investigate the effects of sample orientation on the non-equilibrium dynamics. For this, we consider two orientations: perpendicular (Fig. 3b) and parallel (Fig. 3c) with respect to the direction of gravity. By orientation, we use as a reference the plane at which the heat source is located.

For the sample oriented perpendicular to gravity (Fig. 3b), the intensity maps represent slices along the XY plane (image plane), while the sample depth is parallel to the Z-axis. Both vector components $U_x$ and $U_y$ show identical behavior as expected for a system with radial symmetry. At shallow sample depths, there is a strong flow directed towards the center, i.e., the heat source. The magnitude of this inward flow decreases as a function of sample depth up until 10 μm. At depths exceeding 10 μm, the direction of flow inverts and spreads to a larger area away from the center up until the other liquid-glass interface. The intensity maps for the $U_z$ component as a function of sample depth describe a strong upwards flow in the volume immediately above the heat source that decreases and spreads out with increasing sample depth. Surrounding this volume, there is also a significantly weaker flow in the opposite direction. Ultimately these intensity maps capture the dynamics observed in Supplementary Movie 1, which are primarily dominated by convection and thermo-osmotic flow. Furthermore, experiments probing longer time-scales, show a depletion in particle concentration around the induced temperature field, indicative of thermophobic behavior[31] (Supplementary Movie 2).

Rotating the sample orientation to a parallel configuration leads to a significant change in the dynamics (Supplementary Fig. 6, Supplementary Movie 3), namely, an overall upwards motion of the particles. In this configuration, the intensity maps represent slices along the XZ plane, while the sample depth is directed along the Y-axis (Fig. 3c). For the $U_x$ component, the velocity distribution follows the same trend as in the perpendicular arrangement, with the flow directed towards the heat source at short depth-wise distances away from it, followed by a reversal in direction at depths beyond 10 μm, as expected from thermo-

osmotic flow[18]. The $U_y$ component exhibits a strong flow perpendicularly away from the heat source that decays and spreads with increasing depth. This behavior is similar to $U_z$ in Fig. 3b, yet no buoyancy forces act along this direction and the heat source is stationary, thereby ruling out convection and thermo-viscous flow. Although radiation pressure also acts along this direction, it is not the dominant effect (Supplementary Fig. 5B); thus leaving thermophoresis and thermo-osmosis as the main contributors. The $U_z$ component distribution captures the greatest difference between the two orientations. Namely, at sample depths below 5 μm, the flow is strongly focused towards the heat source analogous to the $U_x$ component; whereas, above 5 μm, an asymmetry develops leading to a strong upwards flow. This upwards flow is dominant across the field of view for depths above 10 μm, and reaches a maximum at 20 μm. Contrary to the perpendicular orientation, this flow extends over an area much larger than the heat source; thereby making the parallel orientation more suitable for fluid actuation.

In summary, the dynamics in the parallel configuration result from the superposition of various competing phenomena (Supplementary Fig. 7), with the three main contributors being: convection, responsible for the overall upwards motion ($U_z$); thermo-osmosis, responsible for the short range in-plane movement of fluid towards the heat source at low sample depths ($U_x$ and $U_z$), which by mass conservation leads to fluid flowing along the optical axis away from the heat source ($U_y$); and thermophoresis, responsible for the short-range movement of particles away from the heat source. Under particular experimental conditions these three phenomena can lead to regions with no net motion of the tracer particles as shown in Supplementary Movie 4, where particles accumulate below the heat source as a result of thermophoresis counteracting out the convection and thermo-osmosis contributions.

The change in orientation of the sample, from perpendicular to parallel, drastically modifies the aspect ratio of the system by more than four orders of magnitude (from 0.006 to 160, respectively), and consequently the dynamics. Here, the aspect ratio is defined as the length of the flow cell system along the Z-axis over the length along the Y-axis. Moreover, despite the comparable microscopic size of the heat source, the water volume involved in the convective dynamics is much larger in the case of the parallel orientation. This phenomenon, in the case of uniformly heated interfaces, has been observed in the seminal works from Braun and Mast et al.[32–34]; nonetheless, there are key differences in the system as a whole when localized heat sources are involved (Supplementary Note 1). Also, contrary to previous reports, the upwards motion of the fluid in our work is unidirectional as a consequence of the localized temperature field. As such, we show that a localized heat source is enough to induce long-range fluid dynamics at low Reynold numbers. To verify that this observation is applicable to more biologically relevant media, we performed additional experiments using either phosphate buffer saline solution or cell culture media as the aqueous media, and observed no significant differences in the dynamics (Supplementary Movie 5). We also observed similar dynamics when we swapped the polystyrene tracer particles for cells, with the caveat that the cell dynamics exhibited a significant contribution from sedimentation (Supplementary Movie 6).

**Localized plasmon heating leads to laminar flow.** The greatly enhanced motion along the Z-axis in the parallel sample orientation represents a promising approach for integrated long-range fluid manipulation based on a microscopic thermal perturbation. To better understand the nature of this flow, which is predominantly determined by convection-driven contributions, we

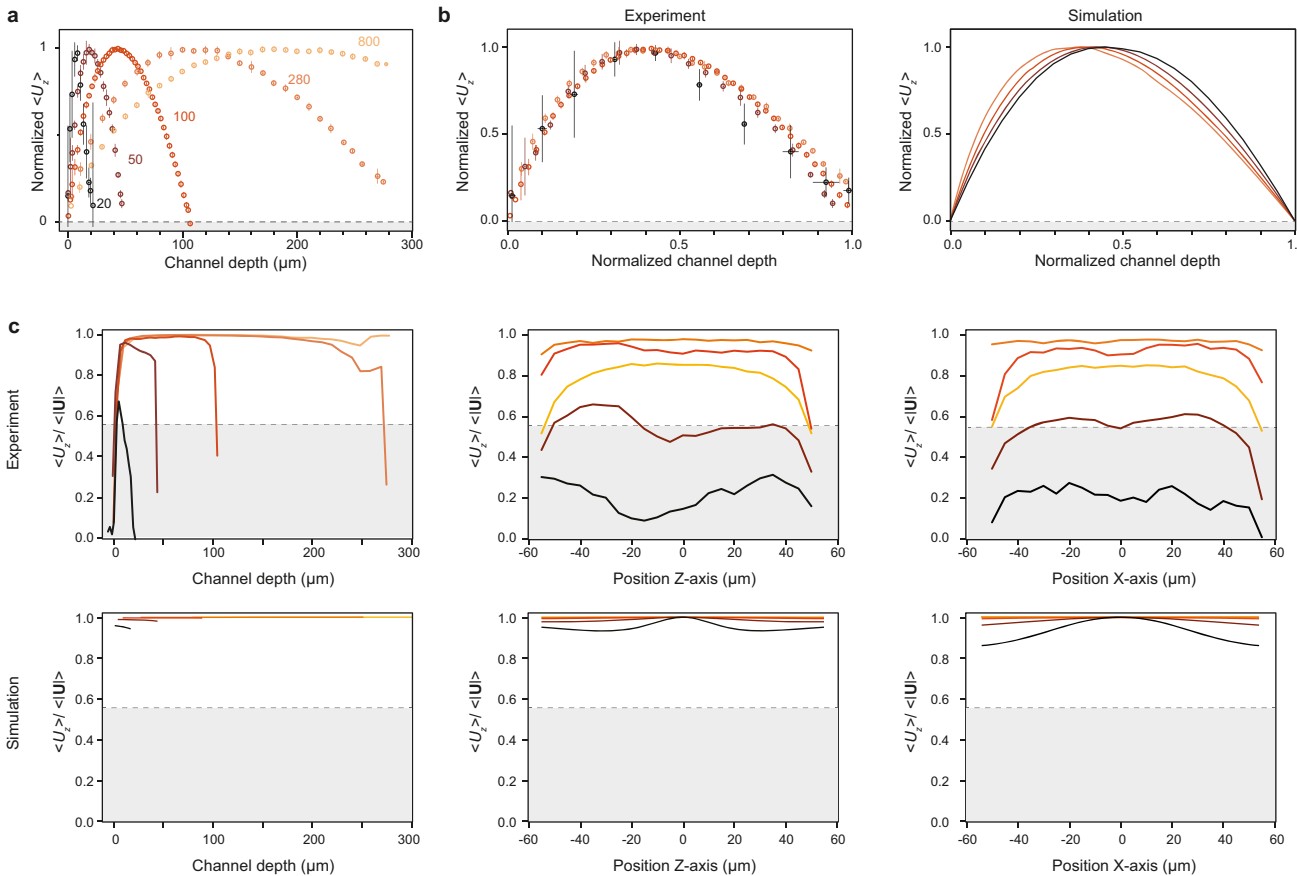

**Fig. 4 Presence of convection-driven laminar flow in the parallel configuration with respect to gravity. a** Normalized flow velocity magnitude for the average Z-component (i.e., parallel to gravity) as a function of channel depth. **b** Same as (**a**) but with a normalized channel depth axis, and simulations results on the right. **c** Empirical figure of merit quantifying the overall contribution of convection to the observed dynamics as a function of channel depth and position along either the x- or z-axis. The shaded area denotes the region above which the laminar flow term becomes dominant. Colors encode the nominal chamber height of each flow cell. Errors bars indicate one standard deviation from the mean of the data.

extract its average value, $<U_z>$, as a function of sample depth for different chamber heights. Specifically, for each sample with a different spacer thickness we: first, perform single-particle-tracking velocimetry experiments; then, compute the corresponding 3D velocity vector field; and finally, determine $<U_z>$ at each given depth position as the mean value across the XZ plane/ intensity map. The chamber height is varied by using silicone spacers of different thickness with nominal sizes ranging from 20 to 800 μm. Only sample depth positions within the first 300 μm are used for this analysis, which are comparable to the working distance of our imaging optics, 500 μm.

In Fig. 4a we show the normalized flow velocities to account for both variations in the induced temperature gradients, and for different flow velocities as a function of channel depth position. Irrespective of chamber height, the flow velocity profile follows a slightly asymmetric paraboloid shape that is characteristic of laminar/Poiseuille flow between two boundaries. By converting the sample depth into a dimensionless parameter via normalization, all velocity profiles from channels smaller than 300 μm overlap within the measurement uncertainty, with a maximum slightly biased towards the heat source (Fig. 4b). This trend can be explained by considering the temperature-dependent viscosity of water, which leads to a lower effective drag force in the proximity of the heat source compared to the other glass-water interface. We verify this flow profile behavior with COMSOL simulations and find good agreement with our results (Methods).

Since fluid convection is not the only active process affecting the non-equilibrium dynamics, we use an empirical figure of merit to quantify its overall contribution as a function of each axis position and channel height (Fig. 4c). This figure of merit corresponds to the ratio of the average upwards component to the average total velocity magnitude ($<U_z>/<|\mathbf{U}|>$). Here, we assume that convection occurs only along the Z-axis and is the dominant contribution in $U_z$. Thus, the higher the figure of merit value, the higher the contributions attributed solely to convection. Figure 4c shows that away from the flow cell boundaries, convection outweighs the other contributions, thus becoming the dominant effect irrespective of chamber height. Furthermore, with increasing chamber height, convection accounts for up to 90% of the motion within the channel, which is reflected in the measurement uncertainty trend present in Fig. 4a–b, i.e., higher uncertainty for decreasing chamber thickness. We also corroborate via simulations that the onset of convection occurs within seconds upon heating (Supplementary Fig. 8).

**Parameter sweep of variables that influence the motion of fluids in the system.** To further characterize the parallel orientation system, we investigate the role of two experimental parameters that are easily tunable: chamber height and heat source size. To do so, we methodically sweep the selected parameter space and determine its effect on the system. To account for variations attributed to different densities of AuNRs at the

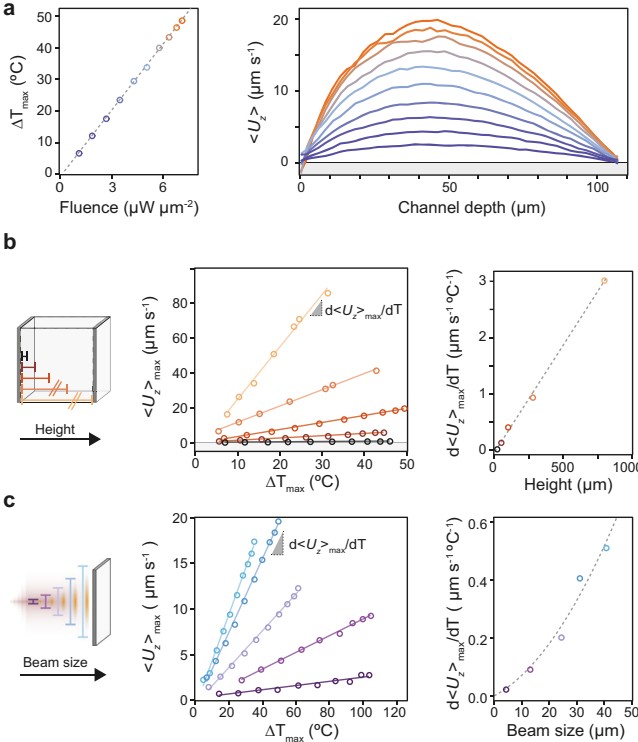

**Fig. 5 Power, chamber height, and beam size dependence of the plasmonically induced convective laminar flow. a** Peak temperature difference as a function of pump fluence (left), and the corresponding plane averaged $U_z$ flow velocity profiles (right) for a sample with a channel height of 100 μm and a pump beam size of 30 μm. **b** Influence of channel height on the maximum laminar flow velocity achieved within the channel for samples with a fixed pump beam size of 30 μm for different peak temperature gradients (left). To determine the dependence between flow velocity and channel depth robustly, i.e., accounting for glass substrate heterogeneity and therefore slightly different temperature profiles between experiments, the slope from each of the previous curves is plotted as a function of channel depth (right). **c** Same as (b) except we probe the influence of pump beam size (heat source) on the maximum flow velocity for samples with a fixed channel height of 100 μm. Dashed and solid lines all correspond to fits to the data. Colors encode the parameter value (fluence, chamber height, and pump beam size) varied during each experiment.

substrate, we perform each parameter sweep as a function of pump fluence and thereby the induced temperature gradient.

By sweeping the pump fluence in a single area, we observe that the induced temperature change scales linearly (Fig. 5a, left panel, Supplementary Movies 7–8). For sake of clarity, we only show the maximum thermal perturbation achieved, which occurs at the middle of the heat source. To determine the effect this has on the fluid dynamics, the average magnitude of the corresponding flow profile, $<U_z>$ is extracted. As shown in the right panel of Fig. 5a, the flow velocity scales with the pump fluence, and by corollary the induced temperature increase. Importantly, the shape of the profile, up to a scaling factor, remains the same irrespective of induced temperature difference. For ease of comparison between parameter sweeps, we use as the read-out parameter the maximum flow velocity along the Z axis denoted as $<U_z>_{max}$.

For the chamber height parameter sweep, we vary the flow cell thickness from 20 to 800 μm, whilst keeping the heat source area fixed at 30 μm (Supplementary Movies 9–10). The plots of maximum temperature increase vs maximum flow velocity confirm a linear relation between these two parameters (Fig. 5b,

left panel). To determine the relation of sample thickness independent of temperature, we compute the slope of each curve in Fig. 5b, denoted as $d<U_z>_{max}/dT$, and plot this value against the experimentally measured chamber height. A linear fit to the data reveals that the flow rate is directly proportional to channel height (Fig. 5b, right panel). This trend fits the model in which the closer the boundaries are to each other, the higher the viscous drag force acting on the liquid, which in turn leads to a lower flower rate.

To determine the role of the heat source size we tune the size of the pump beam from ~4 to 40 μm in diameter (Supplementary Movies 11–12). The chamber height is fixed at 100 μm. We show that given the same maximum temperature at the sample, a larger beam size leads to a higher flow rate (Fig. 5c, left panel). Following the same slope analysis as before, we find that the flow velocity depends quadratically on the size of the heat source. This result agrees with previous theoretical work[12] where the flow velocity is described by the following equation: $U = \frac{L^2 \beta g \Delta T}{\nu}$, where $L$ is the characteristic length of the plasmonic structure (in this case matching the beam size), $\beta$ the thermal expansion coefficient of water, $g$ the gravitational acceleration, and $\nu$ the kinematic viscosity of water. This leads to the important observation that if there is a constraint on the maximum temperature increase a system can experience, which is often the case in lab-on-a-chip applications with temperature-sensitive samples, it is more effective to increase the size of the heat source to achieve higher flow velocities.

**Micro- to mm-scale effects**. The $U_z$ velocity maps from Fig. 3c show that the upwards flow is relatively homogeneous in the XZ-plane away from the boundaries, despite the one order of magnitude smaller cross-section area between the heat source and the field of view. This implies that the buoyancy-driven dynamics extend well beyond the probed area; convection requires by mass conservation that the fluid reverses the direction of flow, as observed in Fig. 3b. Thus, to better probe the length-scale of the convective-dominated dynamics, we reduce the total flow cell area by decreasing the diameter from 8 to 1.5 mm, and record the particle dynamics with two different fields of view. The two fields of view correspond to the digital holographic channel, and a lower magnification standard transmission microscopy one (Methods, Supplementary Movie 13).

In this experiment, a heat source of 19 μm in diameter leads to a maximum increase in temperature of 70 °C in a flow cell with a chamber height of 100 μm (Fig. 6a, left panel). To compare the larger-scale dynamics between the two different fields of view, we apply the concepts of particle imaging velocimetry (PIV) by using the MATLAB PIVlab tool (Methods). In PIV, the fluid velocity vector maps are retrieved by spatially correlating subregions of two subsequent images from a time-lapse recording. The results from the smaller field of view (Fig. 6a, center-left panel), $110 \times 110$ μm², show an even flow velocity, in complete agreement with the previously presented single-particle-tracking velocimetry approach. In the larger field of view, $640 \times 640$ μm², we now observe vorticity in the flow (Fig. 6a, center-right panel). This suggests the presence of two symmetric convection cells that extend beyond the imaged area and towards the edges of the droplet, as confirmed by simulations (Fig. 6a, right panel). Similar results are obtained for different cross-section areas of the flow cell with varying degrees of vorticity (Supplementary Fig. 9). Notably, the mm-scale convective dynamics shown here are achieved with minimal thermal perturbation to the system, as the sample volume experiencing a temperature increase above 1 K is $10^3$ times smaller than the total volume of the system.

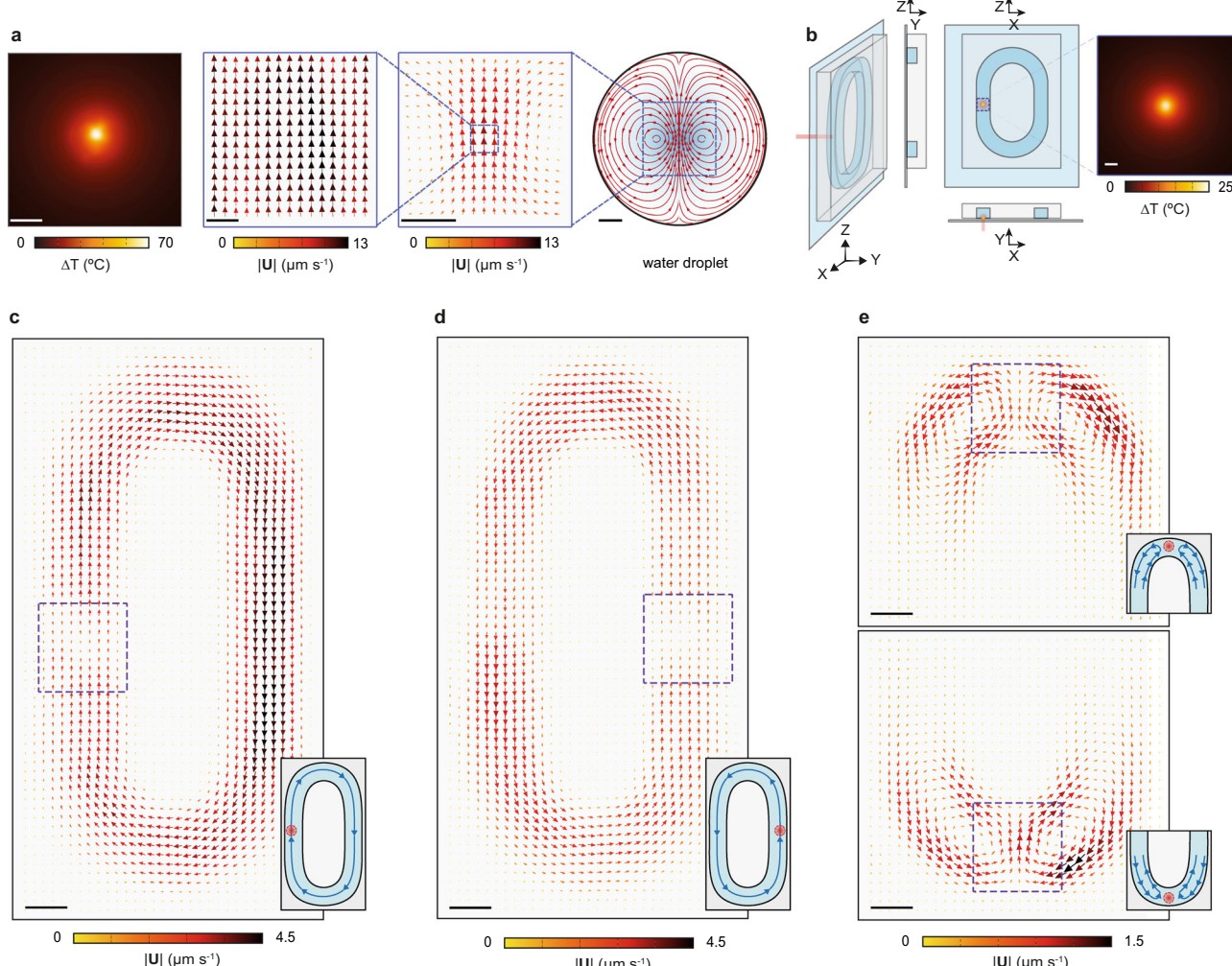

**Fig. 6 Long-range fluid manipulation with a localized thermal gradient. a** PIV-based analysis of the convective flow induced on a 1.5 mm water droplet by a localized thermal gradient. From left to right: measured steady-state temperature increase in the sample (20 μm scale bar), resulting velocity map at high magnification (average of 100 frames collected over 20 s, 20 μm scale bar), and low magnification (average of 100 frames collected over 20 s, 200 μm scale bar), and simulation of the water droplet and induced convection cells (200 μm scale bar). **b** Geometrical representation of the fabricated microfluidic device together with the measured steady-state temperature increase over the dashed area (20 μm scale bar). **c** Induced flow velocity vector field inside the micro channel when plasmonic heating is applied to the left (dashed square, 220 × 220 μm²), **d** right and **e** top and bottom areas of the loop, respectively. (Average of 1000 frames collected over 500 s, 100 μm scale bar).

As a final step, we present an optofluidic strategy to precisely manipulate fluid actuation that is on the one hand reconfigurable, compact, and can be integrated into microfluidic devices; and on the other hand, achieves fine control by minimally perturbing the system with a local heat source. We apply this strategy to a previously reported convection-based microfluidic circuit design[35–37]. The device consists of a PDMS chip containing a closed-loop channel that is then bound to our AuNR-functionalized glass substrate (Fig. 6b). The dimensions of the loop are 0.70 × 1.38 mm² with channel width and height dimensions of 270 × 150 μm², respectively. The concept behind the design is that the dominant buoyancy-driven flow drives the fluid unidirectionally along the loop. The experiments show that a maximum increase in temperature of about 25 °C in the middle of the channel on one side of the loop (Fig. 6c) is enough to induce a steady-state circulation in the loop with a peak velocity of 4.5 μm/s. More importantly in terms of fine-tuned control and reconfigurability, the flow rate can be adjusted optically as demonstrated in Fig. 5a, and the direction of flow can be reversed by illuminating the opposite side of the loop as shown in Fig. 6d

and Supplementary Movie 14. When either the lower or upper portions of the loop are illuminated, the observed dynamics correspond to local convective cells spanning hundreds of microns that lead to no overall fluid circulation (Fig. 6e, Supplementary Movie 15).

## Discussion

In summary, we introduced and validated an optofluidic platform that integrates the capabilities to simultaneously generate and monitor the induced thermal gradient in situ, with the ability to follow single-particle dynamics over a large imaging volume. This approach allowed us to close the gap between reliably producing microscale gradients and characterizing the ensuing non-equilibrium dynamics, thus enabling us to identify the predominant contributing phenomena as a function of different system parameters.

Our experiments demonstrated that localized thermal perturbations at the microscale can lead to mm-scale changes in the dynamics of the system. Here, the most critical parameters

were the orientation of the sample and location of the heat source with respect to gravity, while the chamber height and heat source dimensions, allowed the flow velocity to be finely tuned. By changing the orientation of the sample from perpendicular to parallel, we inherently varied the aspect ratio of the system from low to high. Moreover, the parallel sample configuration is a particularly unique case of a high aspect ratio system because of the localized nature of the heat source and its placement on the sidewall. Furthermore, we highlight that the general dynamics described in our model system also apply to biologically relevant media and objects, thereby making our platform suitable for lab-on-a-chip applications. Our results thus reveal how localized sources of heat can induce complex dynamics and support long-range fluid transport.

Thanks to the multiparameter and multiscale study we explored the properties of these high aspect ratio flow cell systems and developed a strategy that enables all-optical control over a fluid solely from a microscale temperature gradient. The strategy uses a convective loop microfluidic device design arranged in a parallel configuration, whereby the magnitude and direction of the flow are controlled by the fluence of the pump and its position along the loop, respectively.

Beyond the straightforward temperature landscape generated in this work, the colloidal-based plasmonic substrates open the possibility to further engineer the temperature field via beam-shaping[27,28]. This not only allows extensive reconfigurability of the temperature field in space and time, which leads to higher degree of control over the fluid and particle dynamics[15], but also allows greater ease of fabrication compared to nanofabricated plasmonic structures[38]. Furthermore, recent studies have shown that combining electric fields with thermal gradients can significantly enhance long-range transport[19,20], or trap particle ensembles[24,39]. As such, we envision that plasmonic-based heating together with the platform presented in this work will serve as a foundational basis for the development of new technologies that either require fast response times, reconfigurability, or involve the delivery, transfer, or manipulation of temperature-sensitive samples such as proteins, DNA or cells from micro- to mm-length scales.

## Methods

**AuNR preparation**. Gold Chloride trihydrate ($HAuCl_4$), Cetyl trimethylammonium bromide (CTAB), silver nitrate ($AgNO_3$), ascorbic acid, sodium borohydride ($NaBH_4$), polystyrene sulfonate (PSS, Mw 70000), and sodium citrate were purchased from Sigma-Aldrich and used without further purification. AuNRs having a localized surface plasmon resonance (LSPR) absorbance around 780 nm (13 nm × 46 nm) were synthesized according to Nikoobakht et al.[40]. Specifically, a seed solution was prepared in a Teflon vessel by mixing 5 mL of $HAuCl_4$ 0.5 mM with 5 mL of CTAB (200 mM). The mixture was then stirred at 30 °C prior to the addition of 0.6 mL of $NaBH_4$ 10 mM. The resulting fair brown suspension was kept at 30 °C for 3 h before use.

For AuNR growth, a mixture of 50 mL of $HAuCl_4$ (1 mM) and 50 mL of CTAB (200 mM) was continuously stirred at 30 °C. Next, 1.75 mL of $AgNO_3$ (4 mM, 0.68 g/L) and 0.64 mL of ascorbic acid (78.9 mM, 13.9 g/L), were sequentially incorporated into this solution, causing its color to change from brown to lightly faint yellow to colorless in a few minutes. Then, 0.25 mL of the seed solution was added and stirred until a fair pink color developed. Subsequently, 0.36 mL of ascorbic acid solution were added in portions of 0.09 mL every 10 min. The resulting reddish AuNR suspension was kept at 30 °C for 2 h. Exchange of the CTAB capping to citrate was performed as described by Mehtala et al.[41]. Specifically, 40 mL of the AuNR suspension was centrifuged at $22,000 \times g$ over 25 min and 95% (38 mL) of the supernatant was removed. The supernatant was then replaced by 38 mL of a 1.5 mg/mL PSS solution. This processed was repeated twice. After the second centrifugation cycle, the PSS solution was replaced by a 10 mM solution of sodium citrate and followed by three additional centrifugation cycles. Finally, the AuNR suspension in sodium citrate was kept at room temperature prior to use.

**Glass substrate functionalization**. The glass surface facing the objective microscope was functionalized with a uniform coating of AuNR using standard surface functionalization protocols. Briefly, #1.5 borosilicate coverglasses (VWR) were first washed in concentrated aqua regia (1 part nitric acid, $HNO_3$ and 3 parts

hydrochloric acid, HCl), next cleaned in concentrated piranha solution (3 parts sulfuric acid, $H_2SO_4$, and 1 part hydrogen peroxide, $H_2O_2$ 30% vol.), followed by rinsing with milliQ water and later dried. The clean samples were then incubated for 1 h in a 1% w/w solution of APTES (3-aminopropiltriethoxysilane) in ethanol, and subsequently rinsed and sonicated with additional ethanol to remove any excess APTES. Next, the APTES-functionalized substrates were cured for 1 h at 100 °C. For covalent attachment of the AuNRs, the coverslips were immersed in the aqueous nanorod solution for 15 min and gently agitated with a laboratory shaker. Finally, the sample was rinsed with water to remove unbound and excess AuNR, and subsequently submerged in a 1 mg/ml polystyrene PSS aqueous solution to passivate the surface. The density of AuNRs on the substrate was tuned in the range of 20–80 $AuNR/\mu m^2$ for all experiments by varying the concentration of the initial AuNR solution (Supplementary Fig. 1).

**Sample assembly**. To assemble the flow cell sample, we first positioned the silicon gasket on top of the bottom glass, filled the gap with the aqueous solution containing the tracer particles, and finally capped it with the top glass (AuNR functionalized). The silicon spacers self-adhered to the glass substrate and the chamber was sealed by gently applying pressure on the glasses. Silicon gaskets with an inner diameter of 8 mm and supplier-reported thicknesses of 20, 50, and 100 μm from Polymax (1006216, 1006217, and 1006218), and of 250 μm and 1 mm from Grace Bio-Labs (CWS-S-0.25 and JTR8R-A2-1.0) were used. The thickness of the spacers was verified with calipers resulting in values of 20, 55, 105, 280, and 800 μm, respectively. The experimental precision of these measurements was 5 μm. The tracer particles used in all experiments were 1.0 μm polystyrene microspheres (Duke Scientific Corp. ThermoFisher Scientific, R0100). An aqueous solution of tracer particles was prepared by diluting the tracer stock solution in Milli-Q water to the desired concentration. The lowest dilution, $10^2$-fold, was used for experiments with the 20-μm-thick gaskets, whilst the highest, $10^5$-fold, for those with the 800 μm gaskets. Intermediate values of particle dilution were used in all the other cases.

For experiments shown in Supplementary Movie 5, the tracer particles were diluted in either phosphate buffer saline solution or Eagle's minimum essential medium; both at pH = 7.4. For the experiments shown in Supplementary Movie 6, we used human HEK293 cells obtained from American ATCC Cell Line Center (CRL-1573), which were diluted in Eagle's minimum essential medium.

**PDMS microfludics circuit**. The microfluidic circuit presented in Fig. 6 was prepared by binding a polydimethylsiloxane (PDMS) chip on the top of a AuNR-functionalized glass coverslip. The PDMS chip was fabricated using standard soft-lithography techniques. Specifically, a mold was first created by spin-coating a thick layer of photo-resist (Microchem SU8-100) onto a silicon wafer. Next, the microfluidic network pattern was exposed using UV lithography (Quintel Q4000). Then, the PDMS (Sylgard 184) was cast on the mold and subsequently cured during one hour at 80 °C in a convection oven. Once cured, the PDMS was removed from the mold and cut into chips. The channels were inspected with a surface profiler (KLA-Tencor Alpha-Step IQ), confirming a chamber height of 150 μm.

To assemble the microfluidic device, the indentations on the PDMS chip were first filled with the tracer solution and then capped with the glass substrate to create a closed chamber. Care was taken during this process, as sliding the PDMS chip sideways over the glass substrates can cause the AuNR coating to rub-off. To avoid any air gap inside the device, the surface of the PDMS was wetted with a lower surface tension solvent (ethanol) prior to the addition of the aqueous solution. The remaining ethanol inside the channels was replaced with the tracer solution by multiple rinsing steps.

**Optical setup**. We built two almost identical transmission-based pump-probe digital holographic microscope setups arranged in an off-axis configuration that correspond to the perpendicular and parallel orientations, respectively (Supplementary Fig. 10). Both the pump and the probe beams illuminated the sample in a counter-propagating Koehler setup. For the parallel-oriented sample setup, a 465 nm diode laser was used as the probe source (Lasertack LDM-465-3000-c) and a 780 nm diode laser (Lasertack LDM-780-200-c) as the pump. The probe was coupled into a 1 × 2 optical fiber (Thorlabs TW470R5F1) to split the light into the imaging and reference arm of the interferometer. Light from the imaging arm of the interferometer was collected with an Olympus 40x/0.75NA (UPLFLN40XPH) objective, separated from the pump with a 490 nm long pass dichroic mirror (Thorlabs DMLP490L), and focused onto a CMOS camera (Basler acA1920-155um, 5.86 μm × 5.86 μm pixel size) with a 250 mm lens (Thorlabs AC508-250-A), giving a ×55 magnification. A 90:10 beam splitter (Thorlabs BSX16), placed after the 250 mm tube lens allowed the imaging and reference arms to be recombined and interfered at the plane of the camera chip.

The perpendicular-oriented sample setup shared the same features except for the following changes: a 635 nm diode laser (Lasertack LDM-635-200-c) as the probe source, an Olympus 40X/0.65 NA (PLANFL40X) as the imaging objective, a 650 nm long pass dichroic mirror (Thorlabs DMLP650L) to split the pump and probe, and different 1 × 2 optical fiber splitter (Thorlabs TN632R5F1). In both setups the pump source was coupled into the sample by focusing into the back focal

plane of the objective with a 500 mm focal length lens (Thorlabs AC254-500-B). A FPGA card (National Instruments PCI 7831R) was used as a master clock to synchronize the pump and probe lasers together with the camera acquisition.

Data from the microfluidic loops corresponding to Fig. 6a were recorded by adding a secondary lower magnification imaging channel to the parallel setup. A 635 nm laser diode (Lasertack LDM-635-200-c) was used as the illumination source. To lower the magnification of the optical system, three lenses in the following relay system arrangement 150-300-100 mm (Thorlabs AC508-150-A, AC508-300-A and AC508-100-A) were added in the detection path and imaged onto a CMOS camera (Pixelink D755CU, pixel size 3.45 μm × 3.45 μm). The blue (higher magnification) and red (lower magnification) channels were separated using a dichroic mirror (Thorlabs DMLP650L). For the even larger field of view measurements presented in Fig. 6b–e, we used a lower magnification objective: Olympus 20X/0.40 NA (PLN20XPH). The total magnification of the imaging channels after replacing the objective were ×4.95 and ×27.5 for the red and blue channel respectively, as calibrated using a USAF target.

**Optical Imaging**. All the experiments on the model flow cell system (Figs. 2–5) were performed by acquiring 1000 images at a frame rate of either 10 Hz (for channel depths <250 μm) or 100 Hz (for channel depths >250 μm). For the experiments shown in Figs. 6a and c–e the acquisition frame rate was set to 2 Hz, while the number of frames recorded was 100 and 1000, respectively. The sample was only illuminated with the probe pulse during the exposure time of the camera, which was set between 100 and 350 μs. The fluence of the probe was adjusted in the range of $10^{-6}$–$10^{-5}$ mW/μm² corresponding to power at the sample of 60–130 μW for the red (perpendicular configuration and large field of view) and 290–710 μW for the blue (parallel configuration) diode lasers, respectively. The probe in all cases illuminated a circular area with a diameter of 200 μm. The diameter of the beam was determined as equivalent to four sigmas, where the sigma is obtained after fitting the area to a 2D Gaussian. The pump, on the other hand, was switched on after 50 frames had been acquired, and contrary to the probe, continuously irradiated the sample for the duration of the 1000 frame acquisition. The fluence of the pump was adjusted from $6.5 \times 10^{-4}$–$1.8 \times 10^{-1}$ mW/μm² corresponding to a power at the sample ranging from 3.4–22.1 mW and an illumination area between 22–2200 μm². Given at least an order of magnitude lower fluence of the probe compared to the pump, and the orders of magnitude difference in irradiation times, we can rule out any heating contributions attributed to the probe. A minimum of three replicas were acquired for each experimental condition.

**Label-free pump-probe based thermometry**. We used a pump-probe detection scheme to determine the wavefront distortion induced by temperature-dependent refractive-index variations in the aqueous media. We specifically measured this wavefront distortion as an optical phase difference between pump "On" and pump "Off" cycles. For these measurements, the camera acquisition rate was set to 20 Hz, corresponding to a 50 ms frame time. The probe pulse was synchronized with the camera exposure with a duration in the range of 100–200 μs. The pump, on the other hand, was switched on every other camera exposure, with a pulse duration of 49.9 ms and synchronized to end after the probe (Supplementary Fig. 11). For all thermometry experiments reported, a minimum of 100 frames were recorded, corresponding to 50 pump-probe cycles, which were then averaged to increase the signal sensitivity.

In all the experiments reported here, given the weak thermo-optic coefficient of water and the relatively small height of the chamber, the measured phase difference was smaller than 2π, even for temperature changes approaching 100 °C. As a result, there are no issues regarding phase unwrapping that are intrinsic to off-axis digital holography.

From the measured phase differences, we retrieved the three-dimensional temperature map in the sample using the approach in Baffou et al.[30]. In detail, both the measured optical path length difference and temperature distribution are derived from the power density that is absorbed by the sample and subsequently transformed into heat, which we denote as the heat source density (HSD). To obtain the three-dimensional temperature map we first computed the HSD. To do so we converted the measured phase difference map into an optical path length one. The HSD was then obtained by deconvolving the optical path length image with the Greens function that describes the phase distribution from a non-uniform source of heat. Given that image deconvolution is generally an ill-posed inverse problem in the presence of noise, we specifically applied an iterative algorithm based on Tikhonov regularization[42]. The 3D temperature field is then determined by convolving the HSD with the Greens function associated with the Poisson equation.

**Hologram processing**. Recorded off-axis holograms, were processed by first taking a Fourier transformation, which revealed three non-overlapping regions in k-space corresponding to the real, twin, and zero-order images, respectively. Isolation of the real image, corresponding to one of the interference terms, was achieved by hard-aperture selection followed by frequency demodulation, otherwise known as Fourier filtering. Finally, the demodulated real image in k-space was inverse Fourier transformed to yield the desired phase and amplitude images[43].

**Image propagation and 3D SPT**. Holograms were propagated along the optical axis according to the angular spectrum method[43]. Specifically, the processed $M \times M$ pixel² holograms were convolved with a propagation kernel of the form:

$$\mathbf{K}(x, y, z) = \exp\left(iz\sqrt{k_m^2 - k_x^2 - k_y^2}\right)$$

where $k_m = \frac{2\pi n}{\lambda}$, with $n$ being the refractive index of medium through which the light propagates, in this case water. The discretized spatial frequencies are $(k_x, k_y) = 2\pi(x, y)/n\Delta x$ for $(-M/2 \le x, y \le M/2)$ and with $\Delta x$ representing the magnified pixel size of the imaging system. For 3D localization, each hologram was first propagated from $-10$ μm up $+30/70/120/280$ μm with a spacing between different Z-planes ($\Delta z$) of 500 nm, for spacer sizes of 20, 50, 100, 280, and 800 μm; respectively. The resulting 3D intensity maps were then used to find local maxima. To achieve sub-pixel localization in the XY plane, particles that were in focus at a specific Z-plane were fitted by a 2D Gaussian. For the Z-coordinate, sub-$\Delta z$ sampling localization was determined by first calculating the Tamura values $(\mathrm{T}(z) = \sqrt{\frac{\sigma(I_z)}{\mathrm{mean}(I_z)}})$ from the pixel intensities $I_z$ in a region of interest ($\approx 2 \times 2$ μm²) centered about the intensity maxima for each Z-plane, and then fitting a parabola using the two most adjacent pixel values along the maximum[44]. We followed the algorithm of Jaqaman et al. to link the 3D localizations and generate 3D tracks[45]. Only tracks longer than 50 time points were used for further analysis.

**Vector field mapping**. The size of the voxels was varied according to the channel thickness to retrieve the vector field maps. Specifically, the voxel dimensions were only tuned along the optical axis to account for the lower tracer particle concentration and number of measured trajectories associated with increasing channel thickness; corresponding to $5 \times 5 \times 2$ μm³, $5 \times 5 \times 5$ μm³, and $5 \times 5 \times 10$ μm³ for the 20 μm, 50 and 100 μm, and 280 and 800 μm channels, respectively.

**Particle imaging velocimetry**. The velocity maps shown in the paper were calculated using the toolbox PIVlab developed for MATLAB[46]. In this statistical pattern matching technique, every image is divided into small sub-regions, called interrogation areas, from which a displacement vector is computed. The size of the interrogation area determines the resolution of the map. The displacement vectors are obtained by cross-correlation of the interrogation areas from two subsequent images in time. The PIV-derived velocity vector fields reported in Fig. 6 were obtained by setting the interrogation area to $64 \times 64$ pixels² for each image as a good compromise between resolution and visibility. To minimize the influence of Brownian contributions and attain a higher signal to noise ratio the PIV maps were either averaged over 100 frames (Fig. 6a) or 1000 frames (Fig. 6c, d, e).

**COMSOL simulations**. All the simulations present in the paper were computed using COMSOL Multiphysics, making use of the laminar flow, heat transfer in fluids and non-isothermal flow modules. The simulated geometry consisted of a cylinder (5 mm diameter for Supplementary Fig. 8 and 1.5 mm diameter for Fig. 6a) formed by two layers of 170-μm-thick silica glass sandwiching an internal water layer. A symmetry plane bisected the cylinder along its axis in the direction of gravity. Incompressibility and no turbulence conditions were imposed to the fluid laminar flow. Slip conditions were set for the water open surface, accounting for water–air interface, whereas no-slip conditions were set for the water-glass interface. A starting temperature of 25 °C was chosen for the system. A Gaussian beam with a width of 20 μm depositing 1.0 mW power in the form of heat was applied along the cylinder axis at the first glass-water interface. The water open boundaries were set as isolating. External natural air convection from vertical walls was applied to the glass outer surfaces. A thermally open to ambient temperature boundary was assigned for the glass perimetral surfaces. Boussinesq approximation was used in the non-isothermal flow module.

## Data availability

The data produced in this study are available from the corresponding author upon reasonable request.

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

## Acknowledgements

The authors would like to thank Arantxa Albornoz Grados for taking SEM micrographs of the AuNR-functionalized substrates. R.Q., J.O.A., and B.C. acknowledge financial support from the European Research Council program under grants ERC-CoG Qnano-MECA (64790), Fundació Privada Cellex, the CERCA program and the Spanish Ministry of Economy and Competitiveness, under grant FIS2016-80293-R and through the "Severo Ochoa" Program for Centres of Excellence in R&D (SEV-2015-0522).

## Author contributions

R.Q. and J.O.A. conceived the project, designed the experiment and data analysis. J.O.A. developed the data acquisition and analysis software, performed the experiments, and analyzed the data. B.C. prepared the AuNR-functionalized substrates, built the digital holographic microscope, performed the experiments, analyzed the data, and performed COMSOL simulations. J.G.G. fabricated the PDMS microfluidic circuits. I.M. synthesized the AuNR solutions. R.Q., J.O.A., and B.C. wrote the manuscript. All authors read and commented on the manuscript.

## Competing interests

The authors declare no competing interests.
