## [Peer Review File · Nature Communications]

REVIEWER COMMENTS

Reviewer #1 (Remarks to the Author):

Review Ciraulo, Nat Comm 2020

The manuscript by Ciraulo et al. presents a method by which they induce a liquid flow due to the local heating of a thermoplasmonic substrate with gold nanorods, which is in contact with a liquid in a chamber of a few 10 μm in height. The authors measure the temperature distribution inside the fluid chamber as well as the local flow velocity by a 3D holographic tracking of particles. Fluid manipulation techniques that allow a freely configurable manipulation of liquids by means other than a mechanical actuation are powerful and very interesting, and the presented method of convection based fluidics belongs to that category. The new contribution is mainly the simultaneous measurement of 3D fluid flow and temperature distribution. Overall this is a nice experimental demonstration, which might be suitable for Nat. Comm. Before publication, however, I suggest addressing the issues listed below. As convection is a well-understood effect, the authors especially could be more quantitative in comparing their observations to expectations and discussing possible deviations.

- 1) I think the authors should work out in the introductory text the distinction between different thermally driven effects. As mentioned in the manuscript, different groups have shown in the past various types of motions driven by local heat sources. A part of these publications describes experiments that move objects in liquids, but not the liquid; others do move the liquid. Moving the liquid will always yield a long-range flow in the given geometry and is therefore not surprising.
- 2) The group of Dieter Braun has been working with convective effects as well, including modeling of the thermogravimetric flows. Even though these experiments do not rely on localized heat sources, I think this work needs to be considered when addressing the characterization of convective flows.
- 3) On page 9, the authors state that for a parallel orientation of the sample, the flow along the y-direction is caused by radiation pressure and thermo-viscous flow. According to Weinert et al., the thermo-viscous effects require a moving heat source, which is not valid in this case. How could the thermo-viscous effects then contribute?
- 4) A few sentences below, the authors state about the thermophilic nature of the tracer particles. As the particles should follow the flow (as ideal tracer particles), I would not call them thermophilic, since moving towards the heat source is not a property of the tracers but the flow.
- 5) As the temperature field is known from the experiments, I would ask the authors to at least give an estimate of the thermophoretic velocity of the tracers. Similarly, there should be a way to estimate the velocity due to radiation pressure, which is mentioned to contribute to the dynamics.
- 6) I find the notion parallel and perpendicular a bit confusing as the convective effects rely on the direction of the gravitational acceleration, and I would refer to this axis.
- 7) The authors mention theoretical predictions and also present numerical calculations of the flow in the supplementary information. I would have liked to see at least some of the data in SFig. 7 in comparison to the experimental data in the main text. Similar quantitative comparisons should be possible for the velocity along the substrate to show the scaling of the velocity with the distance from the heat source. I feel Fig. 4 can be condensed to contain this information.

8) It would be nice if the authors could comment on the dynamics, e.g., the speed at which the convective flows built up.

9) It would also be nice to have the scale information for the velocity maps in Figure 3 b and c.

Reviewer #2 (Remarks to the Author):

The authors reported an elegant approach to initiate large scale fluidic motion with a localized heat source by simply changing the orientation of the channel relative to the direction of gravity. The reported approach could be utilized to achieve fluidic pumping across a microfluidic network with controllable flow direction that is achieved by moving the position of the laser heating beam. The technical content of the work is well presented and includes simulation results that support the experimental findings. The work is an important contribution to the field of optofluidics and I believe that it is a good fit for Nature Communications. Below are some comments for the authors to address:

- 1.) The authors should comment on what flow behavior would be expected if the channel was tilted say by 45 degrees instead of being vertical.
- 2.) How is the flow field behavior described in this work using a focused heating source different from the case of a uniformly heated surface under the parallel configuration?

Minor comment:

The authors should include a value for the scale bars in the videos.

Reviewer #3 (Remarks to the Author):

The manuscript presents a carefully constructed experimental and numerical study of multiparameter, multiscale (micro- to mm) thermally driven fluid and particle transport. The authors argue that, unlike previous studies that focus on local fluid/particle manipulation over short ranges, their approach permits control over short and long range. The novelty in their work is two-fold. First, it is the unique combination of a thermoplasmonic platform comprising Au nanorods with a digital holographic microscopy platform, as well as 3D particle tracking, to systematically study the effects of thermal perturbation on fluid dynamics from micron to mm-scales. Secondly, the authors show the effects of orientation of a thermoplasmonic platform, either perpendicular or parallel to gravity, on fluid dynamics. The manuscript is well written and interesting. For consideration of publication the authors should address the comments below.

1. Digital holography, and the use of an off-resonance beam, is used to map the temperature-dependent refractive-index profile of the fluid. However, there is often a modulo- 2π phase ambiguity using such interferometric methods. The authors should explain how they calibrated for this effect, at least in the Methods section.
2. The authors would benefit from better articulating the potential applications of their work. If the work is truly for lab-on-a-chip applications, which is often thought of in a biological context, then it should explain, at least briefly, how it translates to biologically relevant media.
3. Related to comment #2 above, such platforms like the one the authors describe could potentially be really useful for cell manipulation. Thus, the authors should comment on how their platform might perform when applied to cells, which are much larger (> 20 microns) and deformable than the tracer particles used in their study. Can they manipulate cell trajectories?

4. Only water was used for the flow studies in the manuscript. Cell culture media consists of many different types of proteins that must be flowed to cells. The authors should comment on how they believe their system would be applied to other biologically relevant media. This would strengthen the applicability of their work.

REVIEWER COMMENTS: NCOMMS-20-18044

Reviewer #1 (Remarks to the Author):

The manuscript by Ciraulo et al. presents a method by which they induce a liquid flow due to the local heating of a thermoplasmonic substrate with gold nanorods, which is in contact with a liquid in a chamber of a few 10 μm in height. The authors measure the temperature distribution inside the fluid chamber as well as the local flow velocity by a 3D holographic tracking of particles. Fluid manipulation techniques that allow a freely configurable manipulation of liquids by means other than a mechanical actuation are powerful and very interesting, and the presented method of convection based fluidics belongs to that category. The new contribution is mainly the simultaneous measurement of 3D fluid flow and temperature distribution. Overall this is a nice experimental demonstration, which might be suitable for Nat. Comm. Before publication, however, I suggest addressing the issues listed below. As convection is a well-understood effect, the authors especially could be more quantitative in comparing their observations to expectations and discussing possible deviations.

We thank the reviewer for taking the time to evaluate our manuscript and provide insight to improve our work. The issues pointed out are addressed below:

1) I think the authors should work out in the introductory text the distinction between different thermally driven effects. As mentioned in the manuscript, different groups have shown in the past various types of motions driven by local heat sources. A part of these publications describes experiments that move objects in liquids, but not the liquid; others do move the liquid. Moving the liquid will always yield a long-range flow in the given geometry and is therefore not surprising.

We thank the reviewer for this suggestion and we have now clarified this point in the introductory text.

Action taken: We have added the following text to the introduction section:

Thermal gradients, broadly speaking, alter the dynamics of the particles in solution at two distinct levels: at the particle and at the fluid level, respectively. At the particle level, the motion of objects in solution along or away the thermal gradient, thermophoresis, is determined by their interactions with the solvent and leads mostly to short-range motion¹. At the fluid level, thermal gradients can induce long-range motion of particles by either convection or thermoviscous flow.

2) The group of Dieter Braun has been working with convective effects as well, including modeling of the thermogravimetric flows. Even though these experiments do not rely on localized heat sources, I think this work needs to be considered when addressing the characterization of convective flows. We refer the referee to response 2.2 to reviewer 2, where we detail the major differences between locally and uniformly heated systems. Unfortunately given the word-limit constraints of the article, we believe that this discussion is best addressed in the supplementary information.

Action taken: The following paragraph has been changed to make reference to the supplementary note and the following citations have been added:

This phenomena, in the case of uniformly heated interfaces, has been previously observed in the seminal works from Braun and Mast et al³³⁻³⁵; nonetheless, there are key differences in the system as a whole when localized heat sources are involved (Supplementary Note), which to the best of our knowledge have not been reported before.

Corresponding Supplementary Note added to the SI document:

Differences between locally and uniformly heated systems:

Focussing on the model flow cell system presented in our work, the first key difference between locally and uniformly heated systems is the resulting temperature field in the water layer, which determines the resulting fluid and particle dynamics. Here, a critical observation is that the heat diffuses through the surrounding water and glass sidewalls and dissipates to the environment (air) by natural convection from the glass surfaces.

In the case of a locally heated system, despite the low heat exchange rate of the external air convection, it is possible to reach a steady state temperature distribution thanks to the relatively large size of the thermal bath (surrounding glass and water) compared to the heat source. Thus, at steady-state, the temperature at any point X in the sample at a distance r much greater than the size of the heat source is well approximated as T_0/r , where T_0 is the temperature at the heat source. In the case of a uniformly heated system^{2,3}, there is no such additional thermal bath within the flow cell, so heat can only diffuse perpendicularly to the heated surface. Therefore, to obtain a steady state temperature field in the water layer, the system requires a heat sink on opposing sidewall, which can be achieved by either using a material with a very high thermal conductivity⁴ (silicon, sapphire), and/or keeping the surface at a fixed temperature via active cooling. The resulting temperature field will be linear, with the distance from the heated surface all across the sample, i.e. proportional to r .

A second key difference is the nature of flow expected in the case of uniformly heated surfaces. In the parallel orientation, the fluid flows upwards close to the hot surface, and downwards close to the cold surface. In the perpendicular orientation, specifically when the system is heated from the bottom, convection arises from an instability when the Rayleigh number exceeds a critical value⁵. This type of convection goes by the name of Reynard-Bénard convection, and leads to the appearance of local convection cells located throughout the sample. Both these scenarios are very different from locally heated systems, such as those presented in the work.

A third key difference involves the thermal inertia of the system, which determines the cooling and heating dynamics, which in turn affects the particle and fluid dynamics. Namely, for locally heated systems, the thermal inertia is small, leading to faster heating and cooling dynamics, which result in a faster system response. This translates into greater reconfigurability and finer control compared to uniformly heated ones. As shown in Fig. S7, the onset of convection-based flow takes place at most within the first 10 s after heating

A fourth key difference is the contribution from particle specific dynamics, which determine whether particles move along or against the thermal gradient¹. These particle transport mechanisms depend on the temperature gradients, which in the case of locally heated systems are localized around the heat source and tend to be greater compared to the uniformly heated system. These particle specific dynamics are evident close to the heat source as shown in the spatial maps in Figure 3c.

To conclude, although locally and uniformly heated systems can achieve similar long-range fluid actuation, the underlying temperature field distribution, fluid transport mechanisms microscale dynamics, and conditions that lead to such behaviour are very different.

3) On page 9, the authors state that for a parallel orientation of the sample, the flow along the y-direction is caused by radiation pressure and thermo-viscous flow. According to Weinert et al., the thermo-viscous effects require a moving heat source, which is not valid in this case. How could the thermo-viscous effects then contribute?

We had initially considered thermoviscous flow as a possible contributor to the dynamics, if we looked at our system from the frame of reference of the steady state fluid moving upwards due to convection; i.e. the fluid is static and it is the heat source that moves downwards. However, the reviewer is indeed correct in pointing out our misunderstanding of the nature of thermo-viscous flow, which should not be valid along the y-axis. As a result, our phenomenological explanation is missing the dominant effect that contributes to the observed flow along the y-direction. This represents a deviation from the expected convection driven dynamics, which the simulations also fail to account for.

Action taken: We have amended the referring text in page 9 to make the reader aware of this deviation, as follows:

This behavior follows the same trend as U_z in Fig. 3b. yet we cannot account for the major phenomena responsible for it. On the one hand, no buoyancy forces act along this direction and the heat source is stationary, thereby ruling out convection and thermos-viscous flow. On the other hand, although radiation pressure acts along this direction, it is not the dominant effect for the observed dynamics as shown in Supplementary Fig. 5b.

4) A few sentences below, the authors state about the thermophilic nature of the tracer particles. As the particles should follow the flow (as ideal tracer particles), I would not call them thermophilic, since moving towards the heat source is not a property of the tracers but the flow.

Although the tracer particles do indeed follow the flow, this does not imply that they do not exhibit dynamics caused by thermal non-equilibrium conditions. At no moment do we claim that the tracers are ideal. In fact polystyrene particles such as the ones used in this work, have been reported to exhibit either thermophobic or thermophilic behaviour depending on the surface coating-solvent interactions and the temperature of the solvent itself⁶.

To ascertain experimentally whether the particles exhibit any significant contributions from particle-dependent dynamics (not flow) we minimize the convective-driven contribution by making the channel height from the flow cell as small as possible (20 μm) under the parallel orientation. Under these conditions (Figure R1), we observe that the tracer particles migrate towards the heat source at low channel depths (0-5 μm) and especially near the heat source. However, at channel depths beyond 5 microns the directionality switches with the particles migrating away from the heat source. This behaviour is also visible in Figure 3c in the main text, for components U_x and U_z . This contrasts with our fluid dynamics simulations, where the dominant contribution is an overall upwards fluid motion regardless of the sample depth position or distance away from the heat source. A more detailed discussion is reported in response 1.7.

To summarize, upon more careful consideration of our data, we agree with the referee that we cannot claim that the observed particle behaviour is either thermophilic or thermophobic. Nonetheless, what we can indeed claim is that these observations arise from a significant contribution from dynamics at the particle level rather than exclusively at the fluid level.

Fig R1: 3D Particle dynamic characterization upon minimizing convective-flow contributions. Resulting 3D velocity field map for an optofluidic platform with a nominal channel height of 20 μm oriented parallel to the direction of gravity. The scheme indicates the plane along which the maps are calculated. Each row and column in the velocity maps correspond to a different vector component and channel depth position, respectively. Specifically, U_r , refers the radial component upon conversion of cartesian to polar coordinates along the XZ plane, where negative values (blue) indicate motion towards the centre and positive values (orange), indicate motion away from the centre. Color encodes the direction and magnitude of the flow velocity, whereas the solid lines represent a contour line indicating a change in direction. All velocity vector components share the same magnitude scale, which has been normalized by the maximum temperature increase in the system.

Action taken: The excerpt indicated by the referee has now been modified as follows:

At sample depths below 5 μm , the flow is strongly focused towards the heat source analogous to the U_x component, indicative of significant contributions from particle driven dynamics. To remark, this non-trivial particle behaviour is also present in the perpendicular orientation.

Also we've adjusted the abstract to reflect the above point:

Here we report an innovative optofluidic platform that fulfills this need by combining digital holographic microscopy with state-of-the-art thermoplasmonics allowing us to identify the different contributions from particle-specific dynamics, convection, and radiation pressure.

5) As the temperature field is known from the experiments, I would ask the authors to at least give an estimate of the thermophoretic velocity of the tracers. Similarly, there should be a way to estimate the velocity due to radiation pressure, which is mentioned to contribute to the dynamics. The thermophoretic velocity, u_t , for colloidal particles under a temperature gradient, ∇T , is given as: $u_t = -D_t \nabla T$, where D_t , is the thermophoretic mobility which is related to the Soret coefficient, S_t , as follows $S_t = D_t/D$. From Fig. S2, we extract the $D = 0.52 \mu\text{m}^2\text{s}^{-1}$. On the one hand, in the absence of any fluid motion contributions, one could estimate the thermophoretic velocity from the instantaneous displacements from particle trajectories upon suppressing the Brownian motion contribution as shown in Figure 3a. On the other hand the Soret coefficient can be approximated by determining D_t from the slope of a plot of u_t vs ∇T . In the context of experiments, we decoupled the contributions attributed to the motion of the fluid by minimizing the convective-driven dynamics, i.e. by making the channel height as small as possible (20 μm) under the parallel orientation. Under these conditions, the velocity maps presented in Fig R1, would thus approximate the normalized thermophoretic velocity. Here the normalization is performed with respect to the maximum temperature change induced in the flow cell system. If we consider the data, where induce a maximum temperature difference of 43K, a plot of u_t vs ∇T within the first 0-5 μm away from substrate with the heat source yields Figure R2 with $D_t = -7.5$ and $-3.4 \mu\text{m}^2\text{s}^{-1}\text{K}^{-1}$; and corresponding to $S_t = -14.4$ and -6.5K^{-1} , respectively.

One can immediately notice that such plot is dependent on the channel depth, and if we were to plot additional sample depth positions we would arrive to the same conclusion stated in response 1.4, namely that there is an inversion in the directionality of the particle flow. We believe that our data, taken under the given experimental conditions, does not provide sufficient insight at this stage to extract meaningful Soret coefficients or to assign whether there is a thermophobic or thermophilic behaviour, since the dynamics are the result of multiple contributing phenomena occurring in the presence of thermal gradients.

Fig R2: Thermophoretic velocity and mobility determination. Estimated thermophoretic velocity as a function of induced thermal gradient at two different channel depth positions, obtained from data generating Figure R1.

Regarding the second point of the reviewer's comment, i.e. estimation of the velocity attributed to radiation pressure, we refer the referee to Figure S5 where we experimentally determined such contribution. Here the experiment is performed in the absence of any gold nanorods on the substrate and in a perpendicular geometry, therefore no thermally driven contributions are present. As a result, the only forces acting on the tracer particles are attributed to the pump beam illuminating the sample in a Koehler configuration, namely radiation pressure (Figure 5a, velocity component U_z) and optical gradient (Figure S5a, velocity components U_x/U_y). The average velocity due to radiation pressure is shown in Figure S5B alongside other contributions for comparison. In this experiment, we found the radiation pressure contribution to be approximately $0.7 \mu\text{m/s}$ at an illumination power at the sample of 15 mW .

6) I find the notion parallel and perpendicular a bit confusing as the convective effects rely on the direction of the gravitational acceleration, and I would refer to this axis.

We thank the referee for this comment; however, we would like to point out that in the manuscript we explicitly defined the orientation of the system according to the reviewer's suggestion, specifically:

“For this we consider two orientations: perpendicular (Fig. 3b) and parallel (Fig. 3c) with respect to the direction of gravity. By orientation, we use as a reference the plane at which the heat source is located.”

Action taken:

In Figure 3b and 3c, we have moved the location of the arrow indicating the direction of gravity closer to the cartoon representation to clarify the convention taken in this work.

7) The authors mention theoretical predictions and also present numerical calculations of the flow in the supplementary information. I would have liked to see at least some of the data in SFig. 7 in comparison to the experimental data in the main text. Similar quantitative comparisons should be possible for the velocity along the substrate to show the scaling of the velocity with the distance from the heat source. I feel Fig. 4 can be condensed to contain this information.

Simulations of our system are consistent with the experimental data in the regime where convective flow is the main phenomena driving the underlying dynamics. This is best observed when we compare our empirical figure of merit, ($\langle U_z \rangle / \langle |U| \rangle$), between experiments and simulations. Here we assume that buoyancy driven dynamics is the major component in $\langle U_z \rangle$. Looking closely at the simulations, we notice that $\langle U_z \rangle$ accounts for more than 90% of the observed velocity, regardless of sample depth position and chamber height. This discrepancy between simulations and experiments is not surprising, as it can be explained by the fact that the simulations do not capture the motion of the particles in response to thermal non-equilibrium conditions and repulsive interactions with the substrate; whereas the experiments do. Similarly, if we now plot the figure of merit as a function of the other two axes (x- and z-axis), we again note a discrepancy between simulations and experiments close to the heat source. Nevertheless the mm-scale, long-range fluid actuation, remains comparable amongst simulation and experiments.

In short, the presence of a thermal gradient, gives rise to particle-specific dynamics that make the motion of particles in the fluid much more complex around the heat source as discussed in response 1.2. In our opinion the best way to capture such rich behaviour is through the spatial maps shown in Figure 3c. Furthermore, we would like to highlight that because the flow profile, U_z , is fairly homogenous along the x-z plane; it is thus possible to collapse the dominant flow profile behaviour to the plots shown in figure 4 and 5; which is not the case for the U_x and U_y components. We have focussed our attention to the convective-term because this is the driving mechanism behind long-range transport -- the thesis of our manuscript.

Action taken:

Modified figure 4 and accompanying text in the main document

Since buoyancy-driven convection is not the only active process affecting the non-equilibrium dynamics, we use an empirical figure of merit to quantify its overall contribution as a function of each axis position and channel height (Fig. 4c).

Fig. 4. Presence of convection-driven laminar flow in the parallel configuration with respect to gravity. (a) Normalized flow velocity magnitude for the average Z-component (i.e. parallel to gravity) as a function of channel depth. (b) Same as (a) but with a normalized channel depth axis, and simulations results on the right (c) Empirical figure of merit quantifying the overall contribution of convection to the observed dynamics as a function of channel depth and position along either the x- or z- axis. The shaded area denotes the region above which the laminar flow term becomes dominant. Colors encode the nominal chamber height of each flow cell.

8) It would be nice if the authors could comment on the dynamics, e.g., the speed at which the convective flows built up.

Upon inspection of the videos provided as supplementary information, one can notice that the onset of the convective flow occurs within the first seconds upon heating. Although in principle this information can be obtained experimentally, in practice this requires either the collection of a large amount of data to segment the particle displacements as a function of time bins to obtain an equivalent signal to noise ratio. This arises from the fact that we rely on temporal and spatial averaging of particle displacements to suppress the Brownian contribution from the observed dynamics. As such, to address this comment we rely on simulations, which show excellent agreement with experiments in the regime where convection-driven dynamics dominate; and more importantly enable us to access early time dynamics, which are inaccessible in our system experimentally, given our limited to time resolution of 10 ms, provided by detector's frame rate.

We compare the temporal evolution of the dominant effect in these flow channels, which is the vertical flow, given by convective flow. From the simulations, we determine that the build up of the convective flow occurs within the first seconds after heating, and reaches a steady value within the first 10 s irrespective of the channel height in this work, in agreement with experimental observations.

Fig. R3. Simulated temporal evolution of the convection-driven flow. a) Normalized flow profile due to convection driven dynamics as a function of time for a channel height of 100 μm with a parallel orientation and heat source size of 20 μm . **b)** Normalized temporal evolution of convective flow as a function of time for different channel heights.

Action taken:

We replaced Figure S7 with Fig R3 shown above, and make reference to it in the results section as follows:

We also corroborate via simulations that the onset of convection occurs within seconds upon heating (Supplementary Fig. 7)

9) It would also be nice to have the scale information for the velocity maps in Figure 3 b and c. Given that the experiments that led to Figure 3b and 3c had slightly different absolute temperature profiles, we opted for ease of comparison between different geometries, as stated in the caption, to normalize the corresponding spatial maps by the maximum temperature increase in the system.

Action taken:

We have added a label on the magnitude scale to clarify that the temperature and velocity components are normalized to the maximum temperature change of the system. Furthermore, we have added the missing scale bars for the spatial maps of 3b and 3c.

Reviewer #2 (Remarks to the Author):

The authors reported an elegant approach to initiate large scale fluidic motion with a localized heat source by simply changing the orientation of the channel relative to the direction of gravity. The reported approach could be utilized to achieve fluidic pumping across a microfluidic network with controllable flow direction that is achieved by moving the position of the laser heating beam. The technical content of the work is well presented and includes simulation results that support the experimental findings. The work is an important contribution to the field of optofluidics and I believe that it is a good fit for Nature Communications. Below are some comments for the authors to address:

We thank the reviewer for supporting the publication of our manuscript. The remaining concerns are addressed below:

1.) The authors should comment on what flow behavior would be expected if the channel was tilted say by 45 degrees instead of being vertical.

We have investigated via simulations given their excellent agreement with experimental results, what would occur if the microfluidic circuit was tilted by a certain angle. We focussed on two rotation axes, specifically along the x- and y-axis, which are illustrated in Figure R4. In both cases, the overall direction of flow is preserved for 45-degree rotations, although the magnitude is reduced by about 30%. The latter is to be expected when one decomposes the resulting flow velocities into their respective axis components $\langle U_x, U_y, U_z \rangle$ and note that the predominant flow contribution, buoyancy-driven, is along the z-direction, so $\cos(\pi/4) \cong 0.7$. A further 45-degree rotation in either cases, a total of a 90-degree rotation, leads to the behaviour reported for the perpendicular geometry, i.e. little to no overall fluid circulation.

Fig. R4. Effect of sample rotation and tilting on the fluid dynamics. Simulated flow velocity vector field upon plasmonic heating obtained upon rotation along the x- or y-axis. Scale bars: 100 μm .

2.) How is the flow field behavior described in this work using a focused heating source different from the case of a uniformly heated surface under the parallel configuration?

To address this question, we would first like to elaborate on the key differences that exist between the two systems.

Focussing on the model flow cell system presented in our work, the first key difference between locally and uniformly heated systems is the resulting temperature field in the water layer, which determines the resulting fluid and particle dynamics. Here, a critical observation is that the heat diffuses through the surrounding water and glass sidewalls and dissipates to the environment (air) by natural convection from the glass surfaces.

In the case of a locally heated system, despite the low heat exchange rate of the external air convection, it is possible to reach a steady state temperature distribution thanks to the relatively large size of the thermal bath (surrounding glass and water) compared to the heat source. Thus, at steady-state, the temperature at any point X in the sample at a distance r much greater than the size of the heat source is well approximated as T_0/r , where T_0 , is the temperature at the heat source. In the case of a uniformly heated system^{2,3}, there is no such additional thermal bath within the flow cell, and the heat can only diffuse perpendicularly to the heated surface. Therefore, to obtain a steady state temperature field in the water layer, the system requires a heat sink on opposing sidewall, which can be achieved by either using a material with a very high thermal conductivity⁴ (silicon, sapphire), and/or keeping the surface at a fixed temperature via active cooling. The resulting temperature field will be linear, with the distance from the heated surface all across the sample, i.e. proportional to r .

A second key difference is the nature of flow expected in the case of uniformly heated surfaces. In the parallel orientation, the fluid flows upwards close to the hot surface, and downwards close to the cold surface. In the perpendicular orientation, specifically when the system is heated from the bottom, convection arises from an instability when the Rayleigh number exceeds a critical value⁵. This type of convection goes by the name of Reynard-Bénard convection, and leads to the appearance of local convection cells located throughout the sample. Both these scenarios are very different from locally heated systems such as those presented in the work.

A third key difference involves thermal inertia of the system, which determines the cooling and heating dynamics, which in turn affects the particle and fluid dynamics. Namely, for locally heated systems, the thermal inertia is small, leading to faster heating and cooling dynamics, which result in a faster system response. This translates into greater reconfigurability and finer control compared to uniformly heated ones. As shown in Fig. S7, the onset of convection-based flow, takes place at most within the first 10 s after heating

A fourth key difference is the contribution from particle specific dynamics, which determine whether particles move along or against the thermal gradient¹. These particle transport mechanisms depend on the temperature gradients, which in the case of locally heated systems are localized around the heat source and tend to be greater compared to the uniformly heated system. These particle specific dynamics are evident close to the heat source as shown in the spatial maps in Figure 3c.

To conclude, although locally and uniformly heated systems can achieve similar long-range fluid actuation in the parallel configuration, the underlying temperature field distribution, fluid transport mechanisms microscale dynamics, and conditions that lead to such behaviour are very different.

Action taken: A Supplementary Note detailed the above discussion has been added in the supporting information document.

Minor comment:

The authors should include a value for the scale bars in the videos.

We thank the reviewer for spotting this.

Action taken: The value for the scale bars has now been added in the corresponding video legend.

Reviewer #3 (Remarks to the Author):

The manuscript presents a carefully constructed experimental and numerical study of multiparameter, multiscale (micro- to mm) thermally driven fluid and particle transport. The authors argue that, unlike previous studies that focus on local fluid/particle manipulation over short ranges, their approach permits control over short and long range. The novelty in their work is two-fold. First, it is the unique combination of a thermoplasmonic platform comprising Au nanorods with a digital holographic microscopy platform, as well as 3D particle tracking, to systematically study the effects of thermal perturbation on fluid dynamics from micron to mm-scales. Secondly, the authors show the effects of orientation of a thermoplasmonic platform, either perpendicular or parallel to gravity, on fluid dynamics. The manuscript is well written and interesting. For consideration of publication the authors should address the comments below.

We thank the reviewer for supporting the publication of our manuscript. We address the remaining concerns point-by-point below:

1. Digital holography, and the use of an off-resonance beam, is used to map the temperature-dependent refractive-index profile of the fluid. However, there is often a modulo- 2π phase ambiguity using such interferometric methods. The authors should explain how they calibrated for this effect, at least in the Methods section.

The reviewer is indeed correct in pointing out the intrinsic issue of phase unwrapping associated with interferometric based methods. Nonetheless, these issues are only present when the measured phase difference is greater than 2π . In all the experiments reported in this manuscript, given the weak thermo-optic coefficient of water and the relatively small height of the chamber, we determine the optical phase difference attributed to a thermal perturbation to be below 2π even for temperature changes approaching 100°C , which is consistent with previous works^{7,8}. Therefore, there is no need to calibrate for this effect. In addition, we provide the reviewer with the following plot that compares the maximum phase difference detected as a function of light fluence on the sample, alongside the retrieved temperature increase that was presented in Figure 5a in the main text.

Fig. R5. Temperature retrieval using phase-based measurements. Measured peak optical phase difference as a function of pump fluence (left), and the corresponding peak temperature difference (right) for a sample with a channel height of $100 \mu\text{m}$ and a pump beam size of $30 \mu\text{m}$.

Action taken: We added the following segment into the corresponding methods section.

“In all the experiments reported here, given the weak thermo-optic coefficient of water and the relatively small height of the chamber, the measured phase difference was smaller than 2π , even for temperature changes approaching 100°C . As a result there are no issues regarding phase unwrapping that are intrinsic to off-axis digital holography”

2. The authors would benefit from better articulating the potential applications of their work. If the work is truly for lab-on-a-chip applications, which is often thought of in a biological context, then it should explain, at least briefly, how it translates to biologically relevant media.

We appreciate the referee for pointing this out and allowing us to improve the visibility and applicability of our work.

In general, by using more complex media, such as biologically relevant ones, one is bound to introduce additional contributions affecting mass and fluid transport in thermal non-equilibrium systems. As a result, the particle and fluid dynamics become more complex, and there are additional parameters available to tune the system¹. For instance, the presence of ions with different thermal diffusivities could lead to the optothermoelectric effect recently reported⁹, whereby a local electric field is induced by charge separation of these ions in solution. This in turn leads to enhanced particle motion for electrically charged species in solution.

Irrespective of how complex the system may be, the platform and analytical tools presented here, allow a detailed characterization of the thermal non-equilibrium dynamics of the system based on relatively straightforward and short experiments. This on the one hand enables to study systems that may be too complex or too computationally demanding to simulate. On the other hand, it allows to fully account for experimental parameters that are often ignored in simulations such as radiation pressure, sedimentation, and in situ temperature profile.

In this work, we mainly focussed our attention to a simple model system to appeal to the broad readership of the journal. Yet, in response to the referee's suggestion, we performed additional experiments using either a phosphate buffer saline (PBS) solution or cell culture media as the aqueous media using polystyrene tracer particles; and cells rather than polystyrene tracer particles. In these experiments, we observed no significant differences in the dynamics of the tracers particles to those obtained with a just water. Likewise, the cells behaved similarly to the tracer particles with the exception that there is a considerable contribution due to sedimentation. Together these observations pave the way towards lab-on-a-chip applications, as the conclusions derived from the model water-based system can be equally extended to biologically relevant media.

Furthermore we refer the reviewer to a list of possible applications stated at the end of the discussion:

“As such, we envision that plasmonic-based heating together with the platform presented in this work will serve as a foundational basis for the development of new technologies that either require fast response times, high and rapid reconfigurability, or involve the delivery, transfer or manipulation of temperature sensitive samples such as proteins, DNA or cells from micro- to mm- length scales.”

Action taken: We added the following segments to the main text:

Results section: Fluid dynamics characterization and influence of sample orientation with respect to gravity

To verify that this observation is also applicable to more biologically-relevant media, we performed additional experiments using either phosphate buffer saline solution or cell culture media as the aqueous media, and observed no significant differences in the dynamics of the tracer particles (Supplementary Movie S3). We also observe a similar behavior if we swap the polystyrene tracer particles for cells, with the caveat that the cell dynamics exhibit a significant sedimentation contribution (Supplementary Movie S4).

Discussion section:

Furthermore, we highlight that the general dynamics studied in detail for our model system also apply to biological compatible aqueous media and systems, thereby making our platform suitable for lab-on-a-chip applications.

Materials and methods section:

For experiments shown in Movie S3, the tracer particles were diluted in either phosphate buffer saline solution or Eagle's minimum essential medium; both at pH=7.4. For the experiments shown in Movie S4, we used human HEK293 cells obtained from American ATCC Cell Line Center (CRL-1573), which were diluted in Eagle's minimum essential medium.

Added Supplementary Movie S3 and S4, and adjusted the ordering of the remaining movies accordingly.

Movie S3. Effect of different aqueous media on the fluid dynamics for a sample orientated parallel to gravity and with a depth of 100 μm . Scale bar: 10 μm .

Movie S4. Evidence for hydrodynamic-based cell manipulation with a localized thermal gradient when the sample is orientated parallel to gravity and with a depth of 100 μm . Scale bar: 10 μm .

3. Related to comment #2 above, such platforms like the one the authors describe could potentially be really useful for cell manipulation. Thus, the authors should comment on how their platform might perform when applied to cells, which are much larger (> 20 microns) and deformable than the tracer particles used in their study. Can they manipulate cell trajectories?

We thank the reviewer for asking this question. As detailed in the response above (response 3.2), we performed experiments in the parallel configuration using cell culture media and cells. In short, we show that the platform can indeed manipulate cells (See Supporting Movie XX). Interestingly, cells given their greater mass, experience a non-negligible sedimentation contribution that acts in the direction of gravity and is thus opposite to the convective-driven flow. As such, depending on the location of the cell with respect to the depth of the flow chamber, its mass and the applied temperature difference, one can either slow down, cancel, or even overcome the sedimentation contribution; thus leading to interesting applications regarding cell manipulation using hydrodynamic forces. However, we would like to stress that a detailed account of the cell manipulation mechanism is beyond the scope of this work, and we consider it better suited if its covered in a separate manuscript, which is in preparation.

Action taken: We added the following segment in the main text along with the supplementary movie:

We also observe a similar behavior if we swap the polystyrene tracer particles for cells, with the caveat that the cell dynamics exhibit a significant sedimentation contribution (Supplementary Movie S4).

4. Only water was used for the flow studies in the manuscript. Cell culture media consists of many different types of proteins that must be flowed to cells. The authors should comment on how they believe their system would be applied to other biologically relevant media. This would strengthen the applicability of their work.

Following the response to the reviewer in 3.2 and 3.3, we followed his suggestion and repeated the experiments for two different aqueous environments that are commonly encountered as biologically-relevant media, PBS and cell culture media; even showcasing the ability to manipulate cells. As stated before, we did not observe any significant differences in flow behaviour. This thereby validates the applicability of our platform to more complex environments, for instance those encountered in biologically relevant media. This is true as long as fluid convection is the dominant transport

mechanism. However, we must note that the dynamics specific either to the different proteins, cells, or more generally to the different chemical species in solution, from either the PBS buffer or the cell culture media were not investigated in detail and lie beyond the scope of this work. We refer the reviewer to the extensive work of Braun's and Mast's group which have investigated the effect of thermal gradients on generating chemical¹⁰ and even pH gradients³. Nonetheless, the tools developed here, would enable any researcher to find the parameters under which convection-driven flow dominates and thus use the strategies for mass and fluid transport discussed in the manuscript.

1. Würger, A. Thermal non-equilibrium transport in colloids. *Reports Prog. Phys.* **73**, (2010).
2. Mast, C. B., Schink, S., Gerland, U. & Braun, D. Escalation of polymerization in a thermal gradient. *Proc. Natl. Acad. Sci. U. S. A.* **110**, 8030–8035 (2013).
3. Keil, L. M. R., Möller, F. M., Kieß, M., Kudella, P. W. & Mast, C. B. Proton gradients and pH oscillations emerge from heat flow at the microscale. *Nat. Commun.* **8**, 1–9 (2017).
4. Duhr, S., Arduini, S. & Braun, D. Thermophoresis of DNA determined by microfluidic fluorescence. *Eur. Phys. J. E* **15**, 277–286 (2004).
5. Roxworthy, B. J., Bhuiya, A. M., Vanka, S. P. & Toussaint, K. C. Understanding and controlling plasmon-induced convection. *Nat. Commun.* **5**, 1–8 (2014).
6. Braibanti, M., Vigolo, D. & Piazza, R. Does thermophoretic mobility depend on particle size? *Phys. Rev. Lett.* **100**, 1–4 (2008).
7. Baffou, G. *et al.* Thermal imaging of nanostructures by quantitative optical phase analysis. *ACS Nano* **6**, 2452–2458 (2012).
8. Goetz, G. *et al.* Interferometric mapping of material properties using thermal perturbation. *Proc. Natl. Acad. Sci. U. S. A.* **115**, E2499–E2508 (2018).
9. Lin, L. *et al.* Opto-thermoelectric nanotweezers. *Nat. Photonics* **12**, 195–201 (2018).
10. Kreysing, M., Keil, L., Lanzmich, S. & Braun, D. Heat flux across an open pore enables the continuous replication and selection of oligonucleotides towards increasing length. *Nat. Chem.* **7**, 203–208 (2015).

REVIEWER COMMENTS

Reviewer #1 (Remarks to the Author):

I thank the authors for their revision and response to my comments and questions. I am actually satisfied with most of the responses and modifications the authors have made.

Yet, I still have a question/comment concerning the response to comment 1.4 and 1.5.

The authors describe in their response the inversion of the direction of motion of the tracer particles with respect to the heat source, when changing the cell height to exclude convective effects. They report the variation of an apparently effective Soret coefficient. For small cell height ($<5 \text{ }\mu\text{m}$), the tracers seem to be attracted to the heat source, while they are repelled if the sample height gets larger than $5 \text{ }\mu\text{m}$. If I understand the observations and reference 18 correctly, then this seems to comply with an interfacial thermo-osmotic flow setting up a flow field in the cell together with the thermophoretic effects. It would be good if the authors consider such contributions in their discussion, especially as they are still missing a contribution and the corresponding paper is actually cited already.

Reviewer #2 (Remarks to the Author):

I thank the authors for addressing my comments. I am satisfied with the revised version of the manuscript.

Reviewer #3 (Remarks to the Author):

The authors have sufficiently responded to my concerns. I recommend the manuscript for publication in Nature Communications.

REVIEWER COMMENTS: NCOMMS-20-18044

Reviewer #1 (Remarks to the Author):

I thank the authors for their revision and response to my comments and questions. I am actually satisfied with most of the responses and modifications the authors have made.

Yet, I still have a question/comment concerning the response to comment 1.4 and 1.5.

The authors describe in their response the inversion of the direction of motion of the tracer particles with respect to the heat source, when changing the cell height to exclude convective effects. They report the variation of an apparently effective Soret coefficient. For small cell height ($<5 \mu\text{m}$), the tracers seem to be attracted to the heat source, while they are repelled if the sample height get larger than $5 \mu\text{m}$. If I understand the observations and reference 18 correctly, then this seems to comply with an interfacial thermo-osmotic flow setting up a flow field in the cell together with the thermophoretic effects. It would be good if the authors consider such contributions in their discussion, especially as they are still missing a contribution and the corresponding paper is actually cited already.

We would really like to thank the reviewer for this comment as it has helped us unravel the different contributing phenomena to the observed dynamics. The reviewer is indeed correct in pointing out the presence of thermo-osmotic flow contributions at the glass water-interface, which we had erroneously overlooked in our initial analysis. This phenomena together with a description of the thermophoretic behaviour of the tracer particles used in our assays now allows us to fully explain the observed fluid and particle transport.

To confirm the presence of thermo-osmotic flow, we first determined the thermophoretic nature of the tracer particles. To do so we performed long term assays with thin flow cell channels ($50 \mu\text{m}$) in the perpendicular configuration to highlight whether the concentration of particles around the localized temperature profile increased or decreased, corresponding to either thermophilic or thermophobic behaviour¹. These experiments confirmed that our tracer particles are indeed thermophobic and are presented in the form of a supplementary video (Movie S2). Therefore the in-plane motion of particles towards the heat source in the proximity of the interface can't be assigned to particle driven dynamics. Instead this motion fits the description of thermo-osmotic flow, which stems from presence of a solid boundary (for instance glass water interface) and a thermal gradient; both of which are present in our model system. Specifically our system counts with a glass substrate that has a uniform coating of PSS (see Material and Methods section), a polar, ionic molecule that give rise to an electric double layer in the water, hence creating very similar experimental conditions to that of a bare glass conditions shown in the original reference².

As a consequence of thermo-osmosis, the inwards flow of the liquid towards the heat source must, by mass conservation, lead to outwards flow of the fluid at the centre of the heat source in the perpendicular. This latter contribution accounts for the missing contribution in the U_y component in Figure 3c. Also, both thermo-osmotic and thermophoretic transport mechanisms are expected to be more relevant at lower sample depths because of the closer proximity to the heat source, which results in greater thermal gradients and a reduction in the convection contributions. Our experiments agree with this as evidenced in Fig. 3b and 3c for sample depths below $5 \mu\text{m}$.

To demonstrate that thermo-osmotic flow, convective flow and thermophoresis are competing transport mechanisms at low sample depth, we present a supplementary movie (Movie S4) where the balance of the three contributions leads to an accumulation of particles below the heat source.

Action taken: We have modified the section of *Fluid dynamics characterization influence of sample orientation with respect to gravity* of the main text as follows:

The intensity maps for the U_z component as a function of sample depth describe a strong upwards flow in the volume immediately above the heat source that decreases and spreads out with increasing sample depth. Surrounding this volume, there is also a significantly weaker flow in the opposite direction. Ultimately these intensity maps capture the dynamics observed in Movie S1, which are primarily dominated by convection and thermo-osmotic flow. Furthermore, experiments probing longer time-scales, show a depletion in particle concentration around the induced temperature field, indicative of thermophobic behaviour³ (Movie S2).

Changing the sample orientation to a parallel configuration leads to a significant change in the dynamics (Supplementary Fig. 6, Movie S3), namely, an overall upwards motion of the particles. In this configuration, the intensity maps represent slices along the XZ plane, while the sample depth is directed along the Y-axis (Fig. 3c). For the U_x component, the velocity distribution follows the same trend as in the perpendicular arrangement, with flow towards the heat source at short depth-wise distances away from it, followed by a reversal in direction at depths beyond 10 μm , as expected from thermo-osmotic flow¹⁸. The U_y component exhibits a strong flow perpendicularly away from the heat source that decays and spreads with increasing depth. This behavior is similar to U_z in Fig. 3b., yet no buoyancy forces act along this direction and the heat source is stationary, thereby ruling out convection and thermo-viscous flow. Although radiation pressure also acts along this direction, it is not the dominant effect (Supplementary Fig. 5B); thus leaving thermophoresis and thermo-osmosis as the main contributors. The U_z component distribution captures the greatest difference between the two orientations. Namely, at sample depths below 5 μm , the flow is strongly focused towards the heat source analogous to the U_x component; whereas, above 5 μm , an asymmetry develops leading to a strong upwards flow. This upwards flow dominant across the field of view for depths above 10 μm and reaches a maximum at 20 μm . Contrary to the perpendicular orientation, this flow extends over an area much larger than the heat source; thereby making it a more suitable orientation for fluid actuation.

In summary, the dynamics in the parallel configuration result from the superposition various competing phenomena (Supplementary Fig. 7), with the three main contributors being: convection, responsible for the overall upwards motion (U_z); thermo-osmosis, responsible for the short range in-plane movement of fluid towards the heat source at low sample depths (U_x and U_z), which by mass conservation leads to fluid flowing along the optical axis away from the heat source (U_y); and thermophoresis, responsible for the short range movement of particles away from the heat source. Under particular experimental conditions these three phenomena can lead to regions with no net motion of the tracer particles as shown in Movie S4, where particles accumulate below the heat source as a result of thermophoresis counteracting out the convection and thermo-osmosis contributions.

The following figure (S7) was added to the supplementary information to summarize the different contributing phenomena responsible for the dynamics in our system:

Supplementary Fig. 7. Contributing phenomena to the dynamics of the parallel orientation. Schematic representation of the different thermal and non-thermal induced phenomena responsible for the dynamics observed in the parallel orientation system. Arrows indicate direction of motion attributed to each contribution.

We have added two supplementary movies, Movie S2 and S4, and adjusted the ordering of the remaining movies accordingly.

Movie S2. Thermally driven particle depletion for a sample oriented perpendicular to gravity and with a depth of 50 μm . Scale bar: 10 μm .

Movie S4. Competing transport phenomena at low sample depth for a sample oriented parallel to gravity and with a depth of 20 μm . Scale bar: 10 μm .

Reviewer #2 (Remarks to the Author):

I thank the authors for addressing my comments. I am satisfied with the revised version of the manuscript.

Reviewer #3 (Remarks to the Author):

The authors have sufficiently responded to my concerns. I recommend the manuscript for publication in Nature Communications.

1. Duhr, S., Arduini, S. & Braun, D. Thermophoresis of DNA determined by microfluidic fluorescence. *Eur. Phys. J. E* **15**, 277–286 (2004).
2. Bregulla, A. P., Würger, A., Günther, K., Mertig, M. & Cichos, F. Thermo-Osmotic Flow in Thin Films. *Phys. Rev. Lett.* **116**, 1–5 (2016).
3. Braibanti, M., Vigolo, D. & Piazza, R. Does thermophoretic mobility depend on particle size? *Phys. Rev. Lett.* **100**, 1–4 (2008).

REVIEWERS' COMMENTS

Reviewer #1 (Remarks to the Author):

The authors have sufficiently responded to my questions. I recommend the manuscript in its present form for publication in Nature Communications.